# A single-cell transcriptomic and anatomic atlas of mouse dorsal raphe *Pet1* neurons

Benjamin W Okaty†*, Nikita Sturrock†, Yasmin Escobedo Lozoya, YoonJeung Chang, Rebecca A Senft, Krissy A Lyon, Olga V Alekseyenko, Susan M Dymecki*

Department of Genetics, Harvard Medical School, Boston, United States

**Abstract** Among the brainstem raphe nuclei, the dorsal raphe nucleus (DR) contains the greatest number of *Pet1*-lineage neurons, a predominantly serotonergic group distributed throughout DR subdomains. These neurons collectively regulate diverse physiology and behavior and are often therapeutically targeted to treat affective disorders. Characterizing *Pet1* neuron molecular heterogeneity and relating it to anatomy is vital for understanding DR functional organization, with potential to inform therapeutic separability. Here we use high-throughput and DR subdomain-targeted single-cell transcriptomics and intersectional genetic tools to map molecular and anatomical diversity of DR-*Pet1* neurons. We describe up to fourteen neuron subtypes, many showing biased cell body distributions across the DR. We further show that *P2ry1-Pet1* DR neurons – the most molecularly distinct subtype – possess unique efferent projections and electrophysiological properties. These data complement and extend previous DR characterizations, combining intersectional genetics with multiple transcriptomic modalities to achieve fine-scale molecular and anatomic identification of *Pet1* neuron subtypes.

**\*For correspondence:**
bokaty@genetics.med.harvard.edu (BWO);
dymecki@genetics.med.harvard.edu (SMD)

†These authors contributed equally to this work

**Competing interests:** The authors declare that no competing interests exist.

## Introduction

Brainstem neurons that synthesize the monoamine neurotransmitter serotonin (5-hydroxytryptamine, 5-HT) (*Baker et al., 1991a*; *Baker et al., 1991b*; *Baker et al., 1990*; *Dahlstroem and Fuxe, 1964*; *Ishimura et al., 1988*; *Steinbusch, 1981*; *Steinbusch et al., 1978*) derive from embryonic precursors that express the transcription factor PET1 (alias FEV) upon terminal cell division (*Hendricks et al., 1999*). PET1 shapes the serotonergic identity of neurons by regulating expression of genes required for 5-HT biosynthesis, packaging in synaptic vesicles, reuptake, and metabolism (*Hendricks et al., 2003*; *Krueger and Deneris, 2008*; *Liu et al., 2010*; *Wyler et al., 2015*; *Wyler et al., 2016*), though some *Pet1*-lineage cells in the brain have ambiguous phenotypes with respect to their ability to synthesize and release 5-HT (*Alonso et al., 2013*; *Barrett et al., 2016*; *Okaty et al., 2015*; *Pelosi et al., 2014*; *Sos et al., 2017*). Aside from shared expression of 5-HT marker genes (to varying degrees), *Pet1*-lineage neurons display wide-ranging phenotypic heterogeneity, including diverse brainstem anatomy, hodology, and expression of neurotransmitters in addition to or other than 5-HT, suggestive of distinct *Pet1* neuron subtypes with divergent neural circuit functions (recently reviewed in *Okaty et al., 2019*). We have previously shown that the mature molecular identities of *Pet1*-lineage neurons strongly correlate with both the embryonic progenitor domain (rhombomeric domain) from which they derive and with their mature anatomy (*Jensen et al., 2008*; *Okaty et al., 2015*), largely consistent with (*Alonso et al., 2013*). However, even within a given *Pet1* rhombomeric sublineage and anatomical subdomain, *Pet1* neurons may display different molecular and cellular phenotypes (*Niederkofler et al., 2016*; *Okaty et al., 2015*). *Pet1* neurons project widely throughout the brain and are functionally implicated in numerous life-sustaining biological processes and human pathologies. Thus, assembling a taxonomy of *Pet1* neuron subtypes based on molecular and cellular properties and linking identified *Pet1* neuron subtypes to specific biological functions is

important for basic neuroscience and human health, including the development of targeted thera-peutics. Here we provide a high-resolution, single-cell transcriptomic atlas of dorsal raphe *Pet1*-line-age neurons revealing hierarchically and spatially organized molecular subtypes, each expressing unique repertoires of neurotransmitters, plasma membrane receptors, ion channels, cell adhesion molecules, and other gene categories important for specifying neuronal functions.

The dorsal raphe (DR) nucleus comprises the largest anatomically defined subgroup of *Pet1* expressing cells in the brain, and these cells are derived from embryonic progenitors in the isthmus and rhombomere 1 (*Alonso et al., 2013*; *Jensen et al., 2008*). Multiple studies have demonstrated neuronal diversity within the DR, in *Pet1*-expressing 5-HT neurons as well as other resident cell popu-lations (*Calizo et al., 2011*; *Challis et al., 2013*; *Crawford et al., 2010*; *Fernandez et al., 2016*; *Huang et al., 2019*; *Kirby et al., 2003*; *Niederkofler et al., 2016*; *Prouty et al., 2017*; *Ren et al., 2018*; *Ren et al., 2019*; *Spaethling et al., 2014*; *Vasudeva and Waterhouse, 2014*; *Zeisel et al., 2018*). In the present study, we extend these findings by transcriptionally profiling *Pet1*-lineage marked DR neurons using microfluidic cell sorting and droplet-based single-cell RNA sequencing (scRNA-seq). Our particular experimental approach combining intersectional mouse genetics, high-throughput cell-type-specific purification (using the On-chip Sort), and newly improved scRNA-seq library construction chemistry (using the 10X Genomics Chromium Single Cell 3′ v3 kit) allowed us to surpass prior resolution of DR *Pet1* neuron molecular diversity, both in terms of the number of DR *Pet1* cells profiled and the number of transcriptomically distinct *Pet1* neuron subtypes identified. To further characterize the anatomical organization of these molecularly defined *Pet1* neuron subtypes, we used intersectional mouse transgenic tools, crossing *Pet1-Flpe* mice with various subtype-rele-vant Cre-driver mice and dual Flpe- and Cre-responsive fluorescent reporter lines. In addition to per-forming histological analyses of these intersectionally defined *Pet1*-lineage neuron subpopulations, we further characterized them using manual cell-sorting from microdissected subdomains of the DR followed by scRNA-seq. Comparing this data with our high-throughput droplet-based scRNA-seq approach allowed us to map *Pet1* neuron molecular diversity onto DR anatomy. We found that DR *Pet1*-lineage neurons comprise as many as fourteen distinct molecularly defined subtypes, several of which we show are anatomically biased within rostral-caudal, dorsal-ventral, and medial-lateral axes. Additionally, by combining intersectional genetics with projection mapping and ex vivo slice electro-physiology we show examples of distinct *Pet1* neuron molecular subtypes that also differ in other cellular phenotypes important for function, such as hodology and electrophysiology.

## Results

### Droplet-based scRNA-seq of *Pet1* fate-mapped DR neurons reveals new molecularly defined neuron subtypes

To characterize the molecular diversity of *Pet1*-lineage DR neurons in a targeted, high-throughput, high-resolution manner we partnered recombinase-based genetic fate mapping, microfluidic fluores-cence-based cell sorting, and droplet-based single-cell barcoding followed by RNA-seq library prep-aration and next-generation sequencing using the 10X Genomics Chromium Single Cell 3′ v3 kit (*Figure 1A*; Materials and methods). Fluorescent labeling of *Pet1*-lineage DR neurons was achieved in mice of the following genotypes: (1) *Tg(Fev-flpe)1Dym* (referred to as *Pet1-Flpe*) (*Jensen et al., 2008*); *En1tm2(cre)wrst* (referred to as *En1-cre*) (*Kimmel et al., 2000*); *GT(ROSA)26Sortm8(CAG-mCherry,-EGFP)Dym* (referred to as RC-FrePe, a dual Flpe- and Cre-dependent fluorescent reporter inserted into the *ROSA26* (*R26*) locus; *Brust et al., 2014*; *Dymecki et al., 2010*; *Okaty et al., 2015*), in which *Pet1*-lineage neurons derived from the *En1*+ isthmus and rhombomere 1 (r1) embryonic progenitor domains are marked by EGFP expression or (2) *Pet1-Flpe; GT(ROSA)26Sortm3.2(Cag-EGFP,CHRM3*/mCherry/Htr2a)Pjen* (referred to as RC-FL-hM3Dq) (*Sciolino et al., 2016*), in which all *Pet1* neurons are EGFP-labeled (Cre was not utilized in these experiments, thus only EGFP, not hM3Dq, was expressed).

Brains were acutely dissected from 6- to 10-week old mice of both genotypes (4 males and 6 females), and DR cells were dissociated as previously described (*Okaty et al., 2015*) (also see Materials and methods). EGFP-expressing neurons were selectively purified using the On-chip Sort (On-chip Biotechnologies Co., Ltd.), a recently developed technology that greatly reduces the pres-sure forces typically exerted on cells in conventional flow sorters, thereby achieving higher levels of

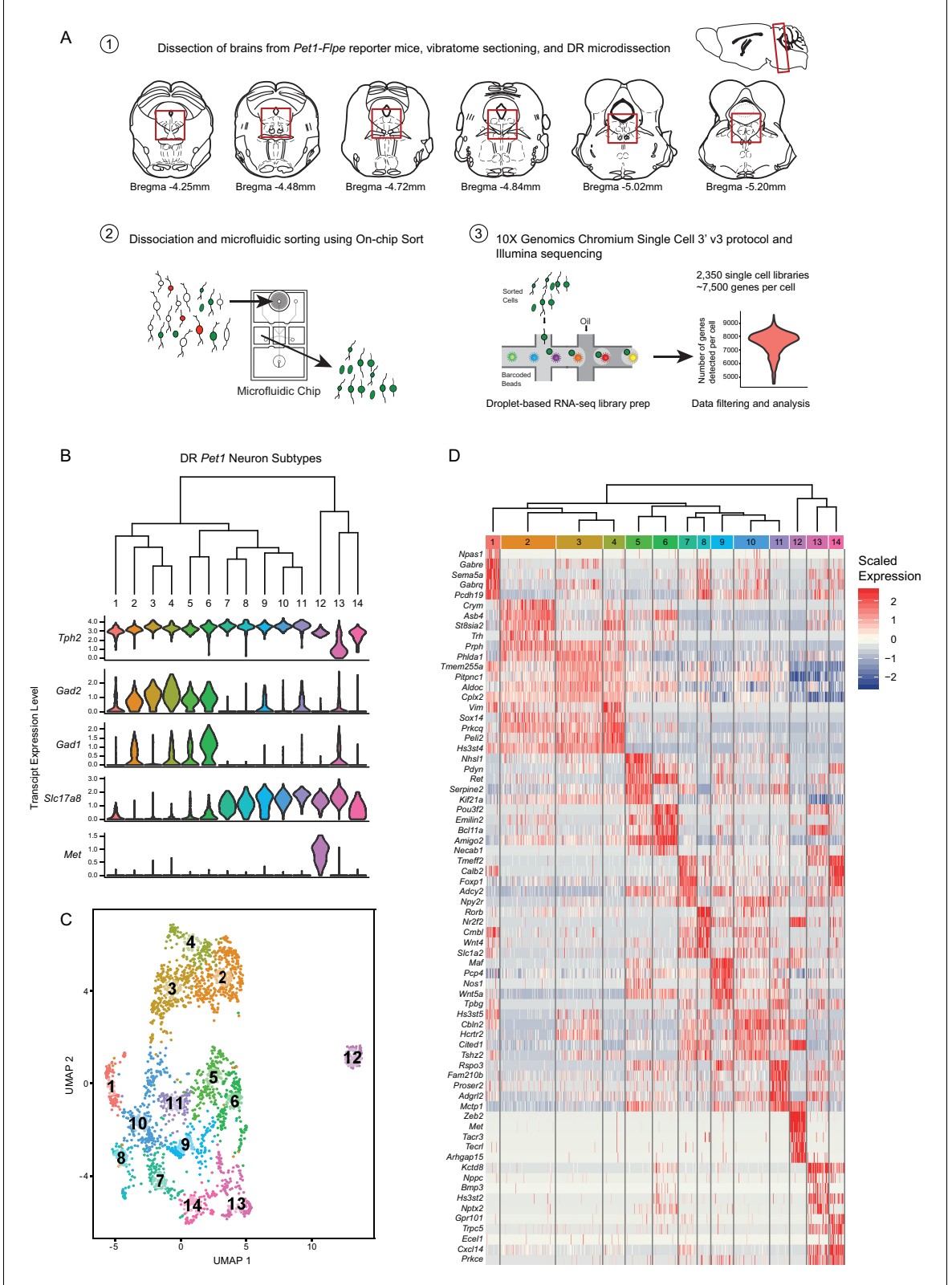

**Figure 1.** High throughput scRNA-seq and clustering analyses reveal as many as fourteen distinct molecularly-defined subtypes (clusters) of *Pet1* neurons in the mouse DR. (**A**) Schematic depicting the experimental and analytical workflow, specifically: (1) brain dissection and DR microdissection, (2) cellular dissociation and microfluidic fluorescence-based cell sorting using the On-chip Sort, and (3) library preparation, sequencing, and analysis using 10X genomics, Illumina sequencing, and the R package Seurat, respectively. (**B**) Hierarchical clustering of *Pet1* neuron subtypes identified by Louvain

*Figure 1 continued on next page*

Figure 1 continued

clustering (using the top two thousand genes with the highest variance, PCs = 1:5, 8:50, and resolution = 0.9), with violin plots depicting the log-normalized expression of a common set of genes (*Tph2, Gad2, Gad1, Slc17a8,* and *Met*) found useful for characterizing cluster structure across multiple resolutions (see *Figure 1—figure supplement 2*). (C) UMAP visualization of single-neuron transcriptome community/similarity structure, with colors and numbers indicating discrete clusters (same clustering parameters as (B)). (D) Heatmap depicting the scaled expression of the top five marker genes for each cluster (ranked by p-value, or in some cases fold enrichment).

The online version of this article includes the following figure supplement(s) for figure 1:

**Figure supplement 1.** Systematic variation of key clustering parameters allows for exploration of the community structure of DR *Pet1* single neuron transcriptomes at variable resolutions.

**Figure supplement 2.** Analysis of clusters at successively increasing resolution values.

**Figure supplement 3.** Expression of serotonin-related genes across DR *Pet1* neuron subtypes.

**Figure supplement 4.** Correlation of 5-HT marker gene expression levels with *Fev* (*Pet1*) gene expression levels for clusters 13 and 14.

cell viability (*Watanabe et al., 2014*). Indeed, examination of sorted neurons revealed that many still had intact processes emanating from their somas. Cells were then run through the 10X Genomics Chromium Single Cell 3' v3 protocol, followed by Illumina NextSeq 500 sequencing. The Cell Ranger pipeline was used for transcript mapping and single-cell de-multiplexing, and all analyses of transcript counts were performed using the R package Seurat (*Butler et al., 2018*; *Stuart et al., 2019*; also see Materials and methods). After stringently filtering out 'suspect' single-cell libraries (e.g. libraries with high levels of non-neuronal or mitochondrial genes, outlier library complexity, or absence/low levels of *Pet1* transcripts), we obtained 2,350 single cells for further analysis, with a mean of 7,521 genes detected per single cell library (mean of 57,678 UMIs per cell), and a total of 17,231 genes detected in aggregate (after filtering out genes that were expressed in fewer than ten cells).

As the principal goal of our scRNA-seq experiments was to characterize molecular diversity of *Pet1*-lineage neurons, our analyses were aimed at identifying genes with significantly variable transcript expression across single neurons, and using these gene expression differences to classify *Pet1*-lineage neuron subtypes. Typical clustering approaches used to classify cell types (or cell states) based on scRNA-seq data are largely unsupervised, but do require supervised input regarding a number of parameters that have the potential to significantly alter the resulting cluster assignments. These parameters include the data reduction used prior to clustering (such as principal components analysis), the number of components included in the reduction, and a resolution or granularity parameter that determines the 'community' size of clusters (i.e. whether cluster boundaries are more or less inclusive; coarse-grained or fine-grained). Rather than arbitrarily choosing a set of parameters for our *Pet1*-lineage neuron subtype classification, we chose a more exploratory approach by systematically varying key parameters and examining how sensitive the resulting clusters were to these combinations of parameters.

First, single-cell transcript counts were log-normalized, and we carried out principal components analysis (PCA) on the scaled and centered expression values of the top two thousand genes with the highest variance (after applying a variance-stabilizing transform, see Materials and methods) in order to reduce the dimensionality of the data onto a smaller set of composite variables that represent the most salient gene expression differences. Plotting the variance explained by each principal component we observed an inflection point, or 'elbow', around the fiftieth component suggesting a drop-off in the information content of subsequent components (*Figure 1—figure supplement 1A*), and found that roughly forty percent of the total variance was explained by these first fifty components. Further examination of the gene loadings of each component revealed that components six and seven were heavily weighted towards sex-specific transcripts and transcripts that largely correlated with batch. As our experiments were not designed to explicitly compare sex as a variable, and given the difficulties of interpreting batch-correlated gene expression differences (e.g. these could stem from population sampling biases of the different cell sorts, different genotypes used, different balance of sexes, or any number of idiosyncratic biological and technical differences) we ultimately chose to remove components six and seven from downstream analyses (though we found that their inclusion had only modest effects on data clustering).

Next, we varied: (1) the number of PCs included in the data reduction (from one to fifty, excluding PCs six and seven) used as input to the Seurat FindNeighbors function, and (2) the resolution

parameter in the Seurat FindClusters function (from 0.1 to 2.0, in intervals of 0.1, using the Louvain algorithm). The results of this analysis are summarized in the heatmap in *Figure 1—figure supplement 1B*. Including only the first few principal components led to highly variable numbers of clusters depending on the resolution parameter (resulting in as many as 40 clusters at the highest resolution). However, for all resolutions the number of clusters mostly stabilized after including the first thirty PCs. In this regime of parameter space the number of clusters was, for the most part, bounded between five and twenty depending on the resolution. As a first pass at homing in on the 'optimum' number of clusters (strictly in a heuristic sense), we calculated the frequency of obtaining a given number of clusters over all combinations of parameters, reasoning that cluster numbers that are less sensitive to precise tuning of parameters would appear more frequently, and the 'robustness' of these cluster numbers might more faithfully reflect biologically meaningful subgroup structure in the data. The cluster number frequency plot in *Figure 1—figure supplement 1C* shows that there are four local maxima and one global maximum corresponding to five, eight, eleven, fourteen, and seventeen clusters respectively. Given the high frequency of these cluster numbers, we chose to examine their composition more carefully. As multiple combinations of parameters lead to the same number of clusters (*Figure 1—figure supplement 1D*), in some cases leading to differences in cluster composition (generally subtle), we decided to err on the side of including more data and thus fixed the number of PCs at one to fifty, excluding PCs six and seven, and varied the resolution to obtain five, eight, eleven, fourteen, and seventeen clusters.

We characterized cluster structure through hierarchical dendrograms, uniform manifold approximation and projection for dimension reduction (UMAP) (a technique recently developed by McInnes, Healy, and Melville as described in a manuscript available at arXiv.org, and implemented in Seurat), and differential expression analysis using Wilcoxon rank-sum tests to identify transcripts that are significantly enriched or depleted among clusters (*Figure 1—figure supplement 2A–I*, *Figure 1B–D*). We ultimately judged seventeen clusters (resolution = 1.5) to be excessive, due to the inclusion of clusters with very few enriched genes as well as clusters that appeared somewhat intermixed in UMAP space (analysis not shown). We thus focused our analyses on lower resolution clusters. Across all resolutions analyzed (0.1, 0.3, 0.7, and 0.9), we found a common set of genes that was useful in characterizing cluster structure, namely *Tph2, Gad2, Gad1, Slc17a8* (alias Vglut3), and *Met*. The *Tph2* gene encodes for tryptophan hydroxylase 2, the rate-limiting enzyme involved in 5-HT biosynthesis in the central nervous system (*Walther and Bader, 2003*; *Walther, 2003*), and all but one cluster showed consistently high *Tph2* transcript expression. In the five- and eight-cluster-number analyses (resolution = 0.1 and 0.3, respectively), one cluster displayed a strikingly bi-modal distribution of *Tph2* transcript expression (*Figure 1—figure supplement 2A,D*, clusters four and six, respectively), however increasing the resolution further divided this group into a *Tph2*-low group (*Figure 1—figure supplement 2G* and *Figure 1B*, clusters eight and thirteen, respectively, corresponding to resolutions of 0.7 and 0.9) and a *Tph2*- 'variable' group, displaying a broader distribution of single-cell expression than other clusters (*Figure 1—figure supplement 2G* and *Figure 1B*, clusters ten and fourteen, respectively). Beyond *Tph2* expression, cluster thirteen (and to a lesser extent cluster fourteen) displayed lower and more variable expression of several 5-HT neuron marker genes (*Figure 1—figure supplement 3*). Interestingly, we found that expression of these genes was significantly correlated with the level of *Pet1* expression in these cells (*Figure 1—figure supplement 4*), consistent with demonstrated transcriptional regulation of 5-HT terminal identity markers by PET1 (*Hendricks et al., 2003*; *Krueger and Deneris, 2008*; *Liu and Deneris, 2011*; *Spencer and Deneris, 2017*; *Wyler et al., 2015*; *Wyler et al., 2016*).

*Gad1* and *Gad2* are paralogous genes that encode two distinct forms of the gamma-aminobutyric acid (GABA) synthetic enzyme glutamate decarboxylase (*Erlander et al., 1991*), and we found a sizable group of *Pet1* neurons (~50%) that express high levels of *Tph2* and *Gad2*, and to a lesser extent *Gad1* (generally detected in fewer cells than *Gad2*) (*Figure 1—figure supplement 2A,B*, clusters one and two), which could be further subdivided into five sub-clusters at finer resolution (*Figure 1B, C*, clusters two-six). One of these clusters, cluster six (*Figure 1B,C*), contained *Pet1* neurons additionally expressing intermediate levels of transcripts for *Slc17a8*, encoding the vesicular glutamate transporter 3 (*Fremeau et al., 2002*; *Gras et al., 2002*). Notably, this group of neurons also had the highest expression of *Gad1* relative to other groups. High expression levels of *Slc17a8* transcripts were detected in ~46% of profiled *Pet1* neurons, comprising eight clusters at finer resolution (*Figure 1B,C*, clusters seven to fourteen), including the *Tph2*-low and *Tph2*-variable clusters

(*Figure 1B,C*, clusters thirteen and fourteen). The most striking outlier group of *Pet1* neurons (cluster twelve in *Figure 1B,C*) is characterized by high transcript expression of *Tph2*, *Slc17a8,* and *Met,* the latter encoding the MET proto-oncogene (also known as hepatocyte growth factor receptor) (*Iyer et al., 1990*). This group of cells consistently clustered separately from all other groups at all resolutions analyzed (*Figure 1—figure supplement 2*). At the finest resolution of 0.9, the remaining 4% of *Pet1* neurons, comprising cluster one, expressed high levels of *Tph2* transcripts but only sporadically expressed transcripts for *Gad2*, *Gad1*, or *Slc17a8* (*Figure 1B,C*).

Examination of the dendrogram in *Figure 1B* and the UMAP plot in *Figure 1C* (as well as examining the successively parcelled UMAP clusters in *Figure 1—figure supplement 2B,E and H* with increasing resolution) gives a sense of 'relatedness' among clusters. For example, *Gad1/2-Tph2* clusters two to four are more similar to each other than to *Slc17a8-Tph2* clusters, while cluster six, the *Gad1/2-Slc17a8-Tph2* cluster, and cluster five are situated between the other *Gad1/2-Tph2* and *Slc17a8-Tph2* groups. Like cluster twelve, clusters thirteen and fourteen appear as outliers from the other clusters in the dendrogram (*Figure 1B*), but despite showing low and variable expression of the 5-HT neuron marker gene *Tph2*, respectively, they nonetheless cluster more closely to other *Pet1* neurons than do *Met-Slc17a8-Tph2-Pet1* neurons (cluster twelve) in the UMAP plot (*Figure 1C*).

*Met*-expressing *Pet1* neurons have been previously reported in mice, both at the transcript and protein levels, specifically in the caudal DR and the median raphe (MR) (*Kast et al., 2017*; *Okaty et al., 2015*; *Wu and Levitt, 2013*) and more recently (*Huang et al., 2019*; *Ren et al., 2019*). Likewise, *Slc17a8*- and *Gad1/2*-expressing DR *Pet1* neurons have been previously reported in mice and rats, as demonstrated by mRNA in situ, immunocytochemistry, and RNA-seq (*Amilhon et al., 2010*; *Commons, 2009*; *Fu et al., 2010*; *Gagnon and Parent, 2014*; *Gras et al., 2002*; *Herzog et al., 2004*; *Hioki, 2004*; *Hioki et al., 2010*; *Huang et al., 2019*; *Okaty et al., 2015*; *Ren et al., 2018*; *Ren et al., 2019*; *Rood et al., 2014*; *Shikanai et al., 2012*; *Spaethling et al., 2014*; *Voisin et al., 2016*). Consistent with functional expression of VGLUT3 protein (encoded by the gene *Slc17a8*), which allows for filling of synaptic vesicles with the excitatory neurotransmitter glutamate, depolarization-induced glutamate release by DR *Pet1*/5-HT neurons has been demonstrated by a number of groups (*Johnson, 1994*; *Kapoor et al., 2016*; *Liu et al., 2014*; *Sengupta et al., 2017*; *Wang et al., 2019*). Additionally, VGLUT3 is thought to interact with vesicular monoamine transporter two (encoded by *Slc18a2,* alias Vmat2; *Erickson et al., 1992*) to enhance the loading of 5-HT into synaptic vesicles by increasing the pH gradient across vesicular membranes, a process referred to as 'vesicle-filling synergy' (*Amilhon et al., 2010*; *El Mestikawy et al., 2011*; *Münster-Wandowski et al., 2016*). GABA-release by *Pet1* DR neurons, on the other hand, has not been reported, thus the functional consequences of *Gad1* and *Gad2* transcript expression are presently unknown.

## Differentially expressed genes span functional categories relevant to neuronal identity

Scaled expression of the top five marker genes for each cluster (ranked by Bonferroni corrected p-value or in some cases fold enrichment) are represented in the heatmaps in *Figure 1—figure supplement 2C,F,I*, and *Figure 1D*, depending on the cluster resolution. For all further analyses, we chose to focus on the 0.9 resolution clustering, as we felt that these fourteen clusters did the best job of parcelling UMAP space. For example, visually-distinguishable groups of cells, like clusters five and six, clusters ten and eight, and clusters seven and fourteen, are each consolidated into a single cluster at resolution = 0.7. While sharing some similarities, these groups differ in the expression of many genes, to an extent that we felt constituted separate classification as supported by the resolution = 0.9 analysis. To aid interpretation of the functional significance of differentially expressed genes, expression patterns of a subset of significantly variable genes and cluster markers are represented in the dot plots in *Figure 2*, organized by categories of biological function (identified by Gene Ontology annotations and literature searches). These gene categories were selected based on general importance for shaping neuronal functional identity – for example genes that encode transcription factors which broadly regulate molecular phenotypes, as well as genes that encode ion channels, plasma membrane receptors, calcium-binding proteins, kinases, and cell adhesion and axon guidance molecules, which collectively govern neuronal electrophysiology, signal transduction, and synaptic connectivity.

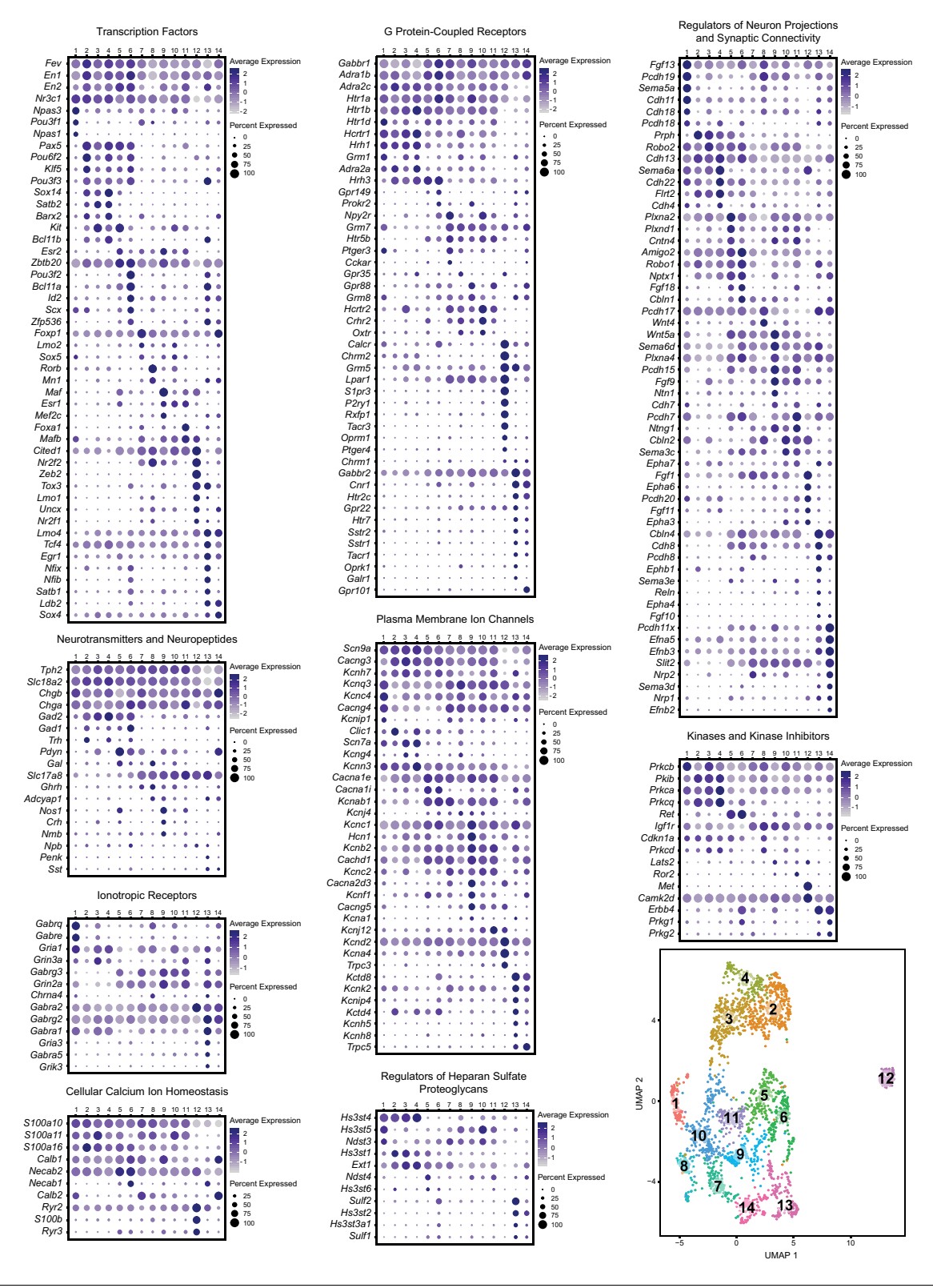

**Figure 2.** Expression patterns of a subset of highly variable genes classified by biological function. Dot plots show the expression of a gene (Y-axis) in each cluster (X-axis), separated by biological function. The size of the dot represents the percentage of cells expressing the gene and saturation of color represents average normalized expression level (scaled and centered). For convenience, the UMAP plot from *Figure 1C* is re-displayed at the bottom right to help link gene expression patterns to overall cluster structure. Minimum inclusion criteria for genes was that they were among the top

*Figure 2 continued on next page*

*Figure 2 continued*

2000 highest variance genes and/or they were found to be significantly enriched or 'de-enriched' in at least one subtype cluster (see Materials and methods).

## Transcription factors

History of expression of *Pet1/Fev,* encoding the FEV transcription factor, ETS family member (*Fyodorov et al., 1998*; *Hendricks et al., 1999*) defines the *Pet1* neuronal lineage. As can be observed from the transcription factor dot plot in *Figure 2* (as well as the violin plot in *Figure 1—figure supplement 3*), *Pet1/Fev* displays broad expression across clusters but is expressed at significantly lower levels in cluster thirteen *Pet1* neurons. Several genes known to be directly regulated by *Pet1* (*Hendricks et al., 2003*; *Liu et al., 2010*; *Wyler et al., 2015*; *Wyler et al., 2016*), such as *Tph2*, *Slc6a4* (Sert) (encoding the serotonin transporter; *Hoffman et al., 1991*; *Lesch et al., 1993*), and *Slc18a2* likewise show reduced expression in cluster thirteen. The transcription factor engrailed 1 (encoded by *En1*) (*Joyner et al., 1985*), in its expression overlap with *Pet1,* is a marker of having derived from progenitors in the isthmus and r1 (*Alonso et al., 2013*; *Jensen et al., 2008*; *Okaty et al., 2015*), and, as expected, *En1* transcripts are detected broadly across all DR clusters. The paralogous gene engrailed 2 (*En2*) (*Joyner and Martin, 1987*), shows a more variable expression profile across *Pet1* DR neurons, being largely absent in cluster eight and twelve, and significantly lower in cluster seven. *En1* and *En2* are required for normal development of DR *Pet1* neuron cytoarchitecture and for perinatal maintenance of serotonergic identity (*Fox and Deneris, 2012*). *Nr3c1*, encoding the nuclear receptor subfamily 3, group C, member 1, aka the glucocorticoid receptor, which binds the stress hormone corticosterone (cortisol in humans), is expressed broadly across clusters one through eleven, but is de-enriched in cluster twelve, and to a lesser extent clusters thirteen and fourteen, suggesting differential sensitivity to corticosterone across different *Pet1* neuron molecular subtypes. Numerous studies have highlighted the functional importance of DR glucocorticoid signaling for 5-HT neuron activity and behavioral modulation (*Bellido et al., 2004*; *Evrard et al., 2006*; *Judge et al., 2004*; *Laaris et al., 1995*; *Vincent et al., 2018*; *Vincent and Jacobson, 2014*).

Other transcription factor encoding genes show more striking expression specificity. Transcripts for neuronal pas domain 1 and 3 (encoded by *Npas1* and *Npas3*) and POU class 3 homeobox 1 (*Pou3f1*) are significantly enriched in cluster one *Pet1* neurons. From mouse genetic studies, both NPAS1 and NPAS3 are associated with regulation of genes and behavioral endophenotypes implicated in psychiatric disorders, such as schizophrenia, though NPAS1/3 are also expressed by other cell types in the brain, such as cortical interneurons, which may contribute to observed behavioral effects of *Npas1/3* loss of function (*Erbel-Sieler et al., 2004*; *Michaelson et al., 2017*; *Stanco et al., 2014*).

*Pax5*, encoding paired box 5 (*Asano and Gruss, 1992*), a transcription factor involved in the regulation of isthmic organizer activity during development (*Funahashi et al., 1999*; *Ye et al., 2001*) is significantly enriched in clusters two, four, and five, and highly expressed in clusters three and six as well. *Pou6f2* (POU class 6 homeobox 2) and *Klf5* (Kruppel like factor 5) show a similar pattern of expression. *Sox14* (SRY-box transcription factor 14) and *Satb2* (SATB homeobox 2) show an even more restricted expression profile, limited to clusters two through four. Notably, clusters two through six are also enriched for expression of *Gad2*, which, like *Sox14*, is most highly expressed in cluster four. *Sox14* expression has been shown to regulate GABAergic cell identity in the dorsal midbrain (*Makrides et al., 2018*), and *Pax5* expression has been implicated in GABAergic neurotransmitter specification in the dorsal horn of the spinal cord (*Pillai et al., 2007*), suggesting that these genes may play similar roles in DR *Pet1* neurons. Interestingly, *Nr2f2* (encoding nuclear receptor subfamily 2, group F, member 2, alias COUP-TFII) shows an expression profile that is complementary to *En2* and *Pax5,* enriched in cluster eight, nine, and twelve, all of which are enriched for *Slc17a8* transcripts, and are largely devoid of *Gad1/2* expression.

Overall, each of the fourteen transcriptome-defined clusters of DR *Pet1* neurons can largely be classified by the combinatorial expression of two to three transcription factors. For example, *Pou3f2* (POU class three homeobox 2), *Bcl11a* (B cell CLL/lymphoma 11A zinc finger protein), and *Id2* (inhibitor of DNA binding 2) show enriched expression in cluster six, and to a lesser extent cluster thirteen.

Other notable transcription factor markers of *Pet1* neuron subgroups include *Foxp1* (forkhead box P1), enriched in clusters seven and fourteen, *Rorβ* (RAR-related orphan receptor beta), enriched in cluster eight, *Maf* (avian musculoaponeurotic fibrosarcoma oncogene homolog), enriched in cluster nine, *Foxa1* (forkhead box A1), enriched in cluster eleven, *Zeb2* (zinc finger E-box binding homeobox 2) enriched in cluster twelve, *Zfp536* (zinc finger protein 536), *Nfix* (nuclear factor I/X), and *Nfib* (nuclear factor I/B), enriched in cluster thirteen (detected in cluster six as well), and *Ldb2* (LIM domain binding 2), enriched in clusters thirteen and fourteen.

## Neurotransmitters and neuropeptides

*Pet1* neuron subtypes defined by transcriptomic clustering also show differential expression of a number of neurotransmitter-related and neuropeptide-encoding genes (*Figure 2* Neurotransmitters and Neuropeptides dot plot). Transcript profiles related to classic neurotransmitter production, including *Tph2, Gad2, Gad1,* and *Slc17a8,* have already been described above (see also *Figure 1B*). Transcript expression of *Trh*, encoding thyrotropin-releasing hormone, is significantly enriched in cluster two *Pet1* neurons and detected in clusters four and six (*Figure 1D* and *Figure 2*). Another gene involved in thyroid hormone signaling, *Crym*, encoding crystalline mu, also known as NADP-regulated thyroid-hormone-binding protein shows a similar expression profile (*Figure 1D*). *Pdyn*, encoding the preprohormone prodynorphin is enriched in clusters five, six, and fourteen. Prodynorphin is the precursor protein to the opioid polypeptide dynorphin, which predominately binds the kappa-opioid receptor to produce a variety of effects, such as analgesia and dysphoria (*Bruchas et al., 2010*; *Chavkin et al., 1982*; *Land et al., 2008*; *Land et al., 2009*). Expression of *Nos1*, encoding nitric oxide synthase 1, is significantly enriched in cluster five, nine, and eleven. The anatomical distribution of nitric oxide expressing DR 5-HT neurons in rodents has been characterized previously as being predominately midline in the DR (*Fu et al., 2010*; *Prouty et al., 2017*; *Vasudeva et al., 2011*; *Vasudeva and Waterhouse, 2014*). Cluster nine also shows enriched expression of *Crh*, encoding corticotropin-releasing hormone. Several other neuropeptide encoding genes show sporadic, significantly variable expression among different clusters, including growth hormone-releasing hormone (*Ghrh*), neuromedin B (*Nmb*), neuropeptide B (*Npb*), proenkephalin (*Penk),* and somatostatin (*Sst).*

## Ionotropic and G protein-coupled receptors

Cluster one and cluster thirteen *Pet1* neurons show the most prominent specificity with respect to ionotropic receptor markers (*Figure 2* Ionotropic Receptors), though in general we found relatively few *Pet1* neuron subtype-specific ionotropic gene markers relative to other categories of gene function. *Gabrq* and *Gabre*, encoding GABA type A receptor subunits theta and epsilon, respectively, are significantly enriched in cluster one, as well as *Gria1*, encoding glutamate ionotropic receptor AMPA type subunit 1. GABA type A receptor subunit gamma3 (*Gabrg3*) and glutamate ionotropic receptor NMDA type subunit 2A (*Grin2a*) transcripts are largely de-enriched in clusters two through four and twelve, are significantly enriched in cluster nine, and variably expressed in other clusters. GABA A receptor subunit alpha 2 (*Gabra2*) is expressed in all clusters but is significantly enriched in cluster twelve, and GABA A receptor subunit alpha 1 (*Gabra1*) and glutamate ionotropic receptor AMPA type subunit 3 (*Gria3*) transcripts both show significant enrichment in cluster thirteen.

Transcripts encoding G protein-coupled receptors (GPCRs) show patterns of enrichment largely across blocks of clusters (e.g. *Slc17a8*-expressing versus non-*Slc17a8*-expressing *Tph2-Pet1* neurons), or highly specific enrichment in either cluster twelve or clusters thirteen and fourteen (*Figure 2* G Protein-Coupled Receptors). For example, cluster twelve neurons show strong enrichment for opioid receptor mu (*Oprm1*), purine receptor y1 (*P2ry1*), relaxin family peptide receptor 1 (*Rxfp1*), sphingosine-1-phosphate receptor *3* (*S1pr3*), and tachykinin receptor 3 (*Tacr3*) transcripts. Moreover, they lack expression of transcripts for many GPCRs expressed by the majority of other *Pet1* neurons, such as presynaptic 5-HT autoreceptors, encoded by *Htr1b* and *Htr1d*, as well as orexin and histamine receptors (e.g. *Hcrtr1, Hcrtr2, Hrh1, Hrh3*), whose protein products are involved in the regulation of arousal. We found that histamine receptor 1 (*Hrh1*) and hypocretin (alias orexin) receptor 1 (*Hcrtr1*) transcripts were the most abundant in clusters one through four, and histamine receptor 3 (*Hrh3*) transcripts were the most abundant in clusters two through six. Hypocretin receptor 2 (*Hcrtr2*) transcripts showed a somewhat complementary expression pattern, with the highest levels in clusters

seven through eleven, as well as cluster three. Other GPCR transcripts with notable expression patterns are neuropeptide Y receptor Y2 (*Npy2r*), enriched in clusters seven and ten, cannabinoid receptor 1 (*Cnr1*) and 5-HT receptor 2C (*Htr2c*), enriched in clusters thirteen and fourteen, and *Gpr101*, an 'orphan' GPCR thought to play a role in the growth hormone releasing-growth hormone signaling axis (GHRH-GH axis) (*Trivellin et al., 2016*; *Trivellin et al., 2018*), enriched in cluster fourteen.

## Regulators of neuron projections, synaptic connectivity, and heparan sulfate proteoglycans

Similar to transcription factor expression patterns, most DR *Pet1* neuron subgroups can be classified by combinatorial enrichment of transcripts for genes encoding regulators of neuron projections and synaptic connectivity (*Figure 2* Regulators of Neuron Projections and Synaptic Connectivity). Differential expression of these genes likely contributes to differential innervation patterns of distinct DR *Pet1* neuron subgroups, such as reported by various studies (*Fernandez et al., 2016*; *Huang et al., 2019*; *Kast et al., 2017*; *Muzerelle et al., 2016*; *Niederkofler et al., 2016*; *Ren et al., 2018*; *Ren et al., 2019*; *Teng et al., 2017*). Genes encoding regulators of heparan sulfate proteoglycans may also play a role in projection specificity and synaptic organization (*Condomitti and de Wit, 2018*; *Di Donato et al., 2018*; *Lázaro-Peña et al., 2018*; *Minge et al., 2017*; *Zhang et al., 2018*), and likewise show patterns of enrichment across different *Pet1* neuron clusters (*Figure 2* Regulators of Heparan Sulfate Proteoglycans). For example, transcript expression of heparan sulfate-glucosamine 3-sulfotransferase 4 (*Hs3st4*) is enriched across clusters one through four, heparan sulfate-glucosamine 3-sulfotransferase 5 (*Hs3st5*) expression is significantly enriched in cluster ten (and expressed at high levels in clusters one, eight, nine, and eleven), and sulfatase 2 (*Sulf2*) and heparan sulfate-glucosamine 3-sulfotransferase 2 (*Hs3st2*) transcripts are enriched in cluster thirteen.

## Intersectional genetic labeling of *Pet1* neuron subgroups in combination with histology and manual scRNA-seq reveals spatial distributions of DR *Pet1* neuron subtypes

Having identified transcriptomically distinct DR *Pet1* neuron subtypes in a largely unsupervised manner, we next sought to determine whether the cell bodies of these molecularly defined *Pet1* neuron subtypes show differential distributions within anatomical subfields of the DR. Using intersectional genetics to fluorescently label *Pet1* neuron subgroups defined by pairwise expression of *Pet1* and one of an assortment of identified subtype marker genes, we iteratively mapped molecular subtypes to anatomy in two ways – (1) using histology and microscopy to directly characterize cell body locations in fixed brain sections (*Figure 3*), and (2) performing manual scRNA-seq on labeled cells dissociated and hand sorted from microdissected anatomical subdomains of the DR, and comparing these expression profiles to our above described high-throughput scRNA-seq data (which we will refer to as our 10X scRNA-seq data) (*Figure 4*). We iteratively bred triple transgenic mice harboring (1) our *Pet1-Flpe* transgene, (2) one of two dual Flpe- and Cre- responsive reporter constructs (RC-FrePe or RC-FL-hM3Dq), and (3) one of five Cre-encoding transgenes (Tg(*Slc6a4-cre*)ET33Gsat (referred to as *Slc6a4-cre*), *Slc17a8*^tm1.1(cre)Hz^ (referred to as *Slc17a8-cre*), *Npy2r*^tm1.1(cre)Lbrl^ (referred to as *Npy2r-cre*), Tg(*Crh-cre*)KN282Gsat/Mmucd (referred to as *Crh-cre*), or *P2ry1*^tm1.1(cre)Lbrl^ (referred to as *P2ry1-cre*), where *cre* expression is driven by either the endogenous promoter of the marker gene or by a gene-specific bacterial artificial chromosome (BAC). In selecting candidate markers from our list of differentially expressed genes, we sought gene drivers that could potentially divide *Pet1* neurons into subgroups at varying resolutions and were available as *cre* lines. Representative images for each triple transgenic genotype are given in *Figure 3* (organized by marker genes, columns A-E, at different rostrocaudal levels of the DR, rows 1–6). For each genotype, the intersectionally defined subpopulation of neurons is labeled in green (i.e. history of Flpe and Cre expression) whereas the 'subtractive' subpopulation is labeled in red (i.e. history of Flpe but not Cre expression).

## Histology of *Pet1*-Intersectionally defined neuron populations

High *Slc6a4* expression, like high *Tph2* expression, defines *Pet1* neuron clusters one through eleven. Cluster twelve shows consistently lower mean expression of *Slc6a4* transcripts (and to a lesser extent *Tph2* transcripts) than clusters one through eleven (*Figure 3A*), cluster fourteen shows a broader

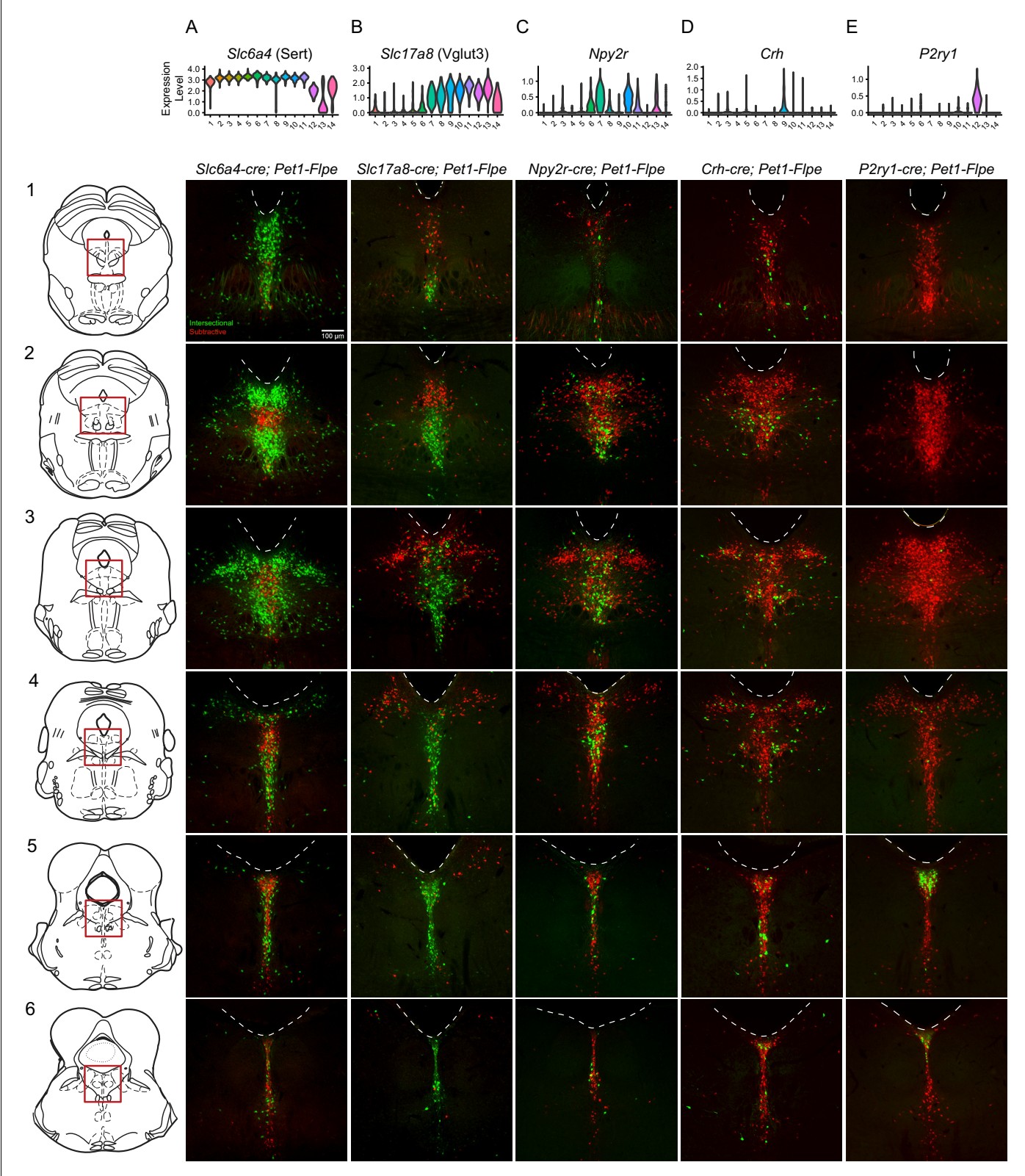

**Figure 3.** Intersectionally targeted *Pet1* neuron subtypes have different anatomical distributions in subregions of the DR. (A–E) Low magnification view of 40 μm coronal sections showing the DR from rostral to caudal (1-6) in triple transgenic animals. Cell bodies are labeled by the intersectional expression of a Cre driver of interest, *Pet1-Flpe*, and the intersectional allele RC-FrePe (green EGFP marked cells expressing both Cre and Flpe and red mCherry expressing *Pet1-Flpe* subtractive population) unless otherwise noted. (A) *Slc6a4-cre; Pet1-Flpe*; RC-FrePe, (B) *Slc17a8-cre; Pet1-Flpe*; RC-FL-

*Figure 3 continued on next page*

Figure 3 continued

hM3Dq (green mCherry-hM3Dq marked cells expressing Cre and Flpe and red EGFP expressing *Pet1-Flpe* subtractive population), (C) *Npy2r-cre; Pet1-Flpe*; RC-FrePe, (D) *P2ry1-cre; Pet1-Flpe*; RC-FrePe. Top row shows violin plots depicting transcript expression (10X scRNA-seq data) of the respective gene corresponding with each Cre driver. Scale bar (A1) equals 100 µm. The expression of TPH2 and VGLUT3 was further investigated in *Figure 3—figure supplements 1* and *2*, respectively.

The online version of this article includes the following figure supplement(s) for figure 3:

**Figure supplement 1.** Diversity of *Tph2* RNA transcripts and protein expression in dorsal raphe *Pet1* neurons.

**Figure supplement 2.** VGLUT3 antibody staining of *Pet1* neurons is anatomically biased within different DR subdomains.

distribution of *Slc6a4* transcript levels (*Figure 3A*) and many 5-HT markers (*Figure 1—figure supplement 3*), and cluster thirteen shows very low levels of *Slc6a4* transcripts (*Figure 3A*) and other 5-HT markers (*Figure 1—figure supplement 3*). Consistent with the majority of profiled *Pet1* neurons expressing high levels of *Slc6a4* and *Pet1* transcripts, we detected intersectional *Slc6a4-cre; Pet1-Flpe* fluorescently marked neurons throughout the full rostrocaudal and dorsoventral extent of the DR (*Figure 3A1–A6*), however the subtractive population (presumably cluster thirteen and perhaps some cluster twelve and fourteen *Pet1* neurons) showed a more limited distribution. These subtractive (Flpe+ but Cre-) neurons were most conspicuously concentrated in the rostromedial DR (*Figure 3A2*), where only a few *Slc6a4-cre; Pet1-Flpe* intersectional (Flpe+ and Cre+) neurons were intermingled. More caudally, the Flpe-only subtractive neurons remained largely midline, but became more intermixed with the double-positive *Slc6a4-cre; Pet1-Flpe* intersectionally marked cells. As another way of anatomically characterizing putative cluster thirteen *Pet1* neurons, we immunostained for TPH2 in *En1-cre; Pet1-Flpe*; RC-FrePe mice (the same genotype as used in some of our 10X scRNA-seq experiments), and found that the distribution of TPH2 immunonegative *Pet1* neuron cell bodies showed a very similar distribution to the subtractive neurons (Flpe-only) in *Slc6a4-cre; Pet1-Flpe*; RC-FrePe mice. (*Figure 3—figure supplement 1A*), further confirming the existence of *Pet1*-expressing neurons that do not express TPH2 protein (*Barrett et al., 2016*; *Pelosi et al., 2014*). To explicitly examine the relationship of *Tph2* transcript level to presence or absence of TPH2 protein, we additionally performed concurrent TPH2 immunostaining and *Tph2* single molecule fluorescent in situ hybridization in DR-containing brain sections of an *En1-cre; Pet1-Flpe*; RC-FrePe mouse (*Figure 3—figure supplement 1B–G*). Similar to our 10X scRNA-seq data, we found a bimodal distribution of *Tph2* transcript abundance, with the majority of single cells distributing in the higher mode (*Figure 3—figure supplement 1D*). Across all subregions of the DR analyzed, we found that dual EGFP and TPH2 immunopositive cells contained significantly more *Tph2* transcripts than EGFP immunopositive TPH2 immunonegative cells (p-value<0.01, Wilcoxon Rank Sum tests with Benjamini and Hochberg correction for multiple comparisons, see Materials and methods, *Figure 3—figure supplement 1E*). However, as the TPH2 immunopositive and negative *Tph2* transcript distributions showed some overlap, cells with low *Tph2* transcript counts occasionally expressed TPH2 protein and cells with higher transcript counts occasionally did not. In particular we found that the third and fourth deciles (the 'transition zone' between modes) of the *Tph2* transcript distribution displayed the greatest degree of intermixing of TPH2 positive and negative cells (*Figure 3—figure supplement 1D*). Intriguingly, we also found that the somata of EGFP positive TPH2 negative cells were significantly smaller than dual EGFP and TPH2 positive cells (p-value<0.01, Wilcoxon Rank Sum tests with Benjamini and Hochberg correction for multiple comparisons, *Figure 3—figure supplement 1F,G*).

*Slc17a8-Pet1* expression defines *Pet1* neuron clusters seven through fourteen, and to a lesser extent cluster six (*Figure 3B*). We observed that *Slc17a8-cre; Pet1-Flpe* intersectionally marked neurons show a strong ventromedial bias in rostral portions of the DR (*Figure 3B1–B3*), and are the predominant *Pet1* neuron subgroup in the more caudal midline DR (*Figure 3B3–B6*). By contrast, the subtractive *Pet1* neuron subgroup (presumably comprising *Pet1* neurons from clusters one through five and partly six) show a strong dorsal and lateral bias and are largely absent from the most caudal portions of the DR. We further characterized VGLUT3 protein expression in *Pet1* neurons by VGLUT3 immunohistology in *Slc17a8-cre; Pet1-Flpe*; RC-FL-hM3Dq mice. We found consistent overlap between intersectional recombination marked neurons and VGLUT3 protein expression, especially in medial, ventromedial, and caudal portions of the DR (*Figure 3—figure supplement 1C–J*). In the

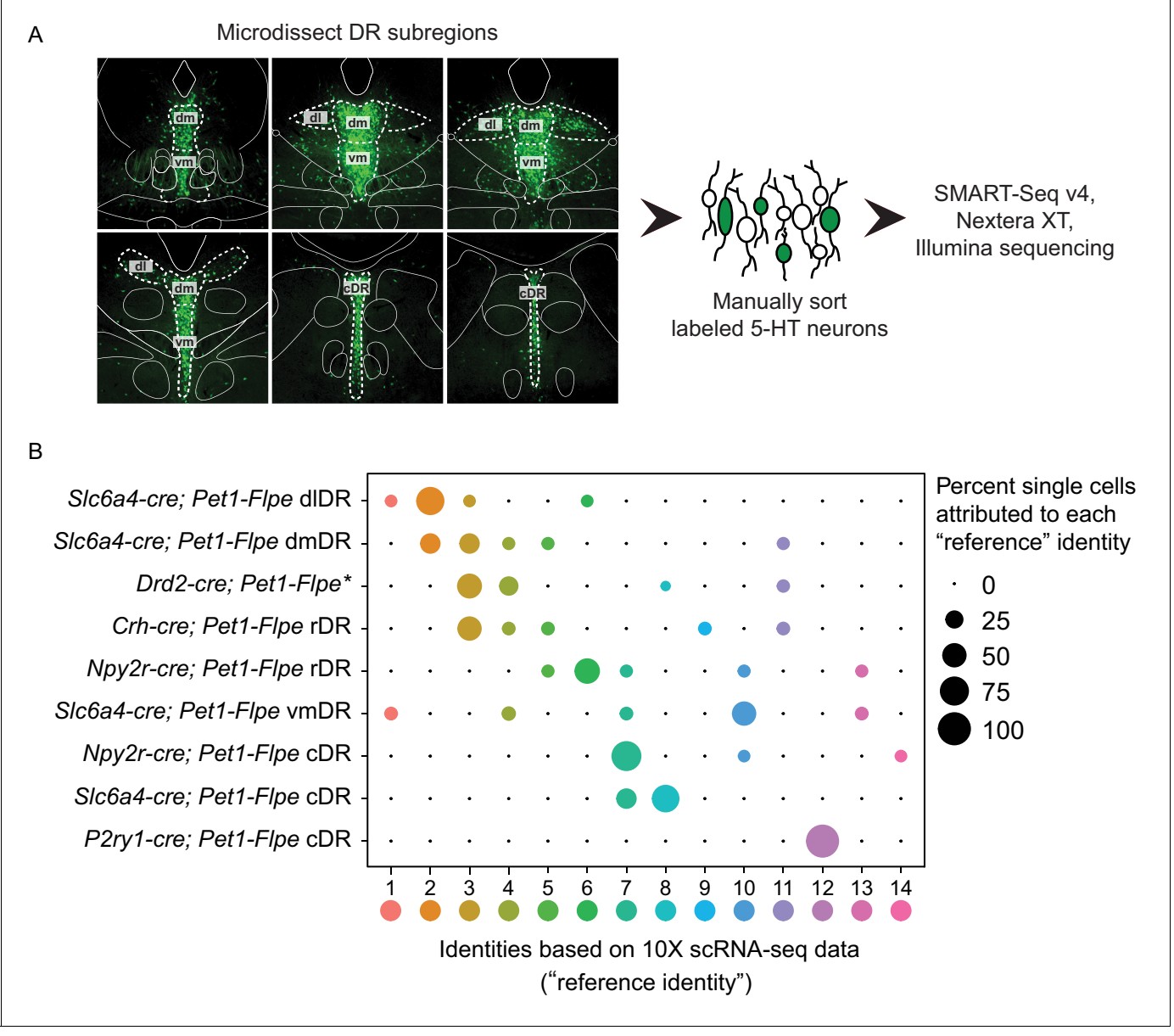

**Figure 4.** scRNA-seq of *Pet1* neurons manually sorted from anatomical subdomains map onto specific 10X scRNA-seq clusters. (**A**) Schematic of the pipeline used for manual sorting and sequencing, including referenced anatomical subdomains mapped onto representative images of the DR. *Pet1* neurons are in green. (**B**) Dot plot mapping manually sorted cells from a given genotype and anatomical subdomain (Y-axis) to the fourteen 10X clusters (X-axis). The size of the dot indicates the percentage of single cells from a genotype/anatomical region attributed to a reference cluster. Note, the asterisks after *Drd2-cre; Pet1-Flpe* is to denote that these data come from a previously published study (*Niederkofler et al., 2016*), and these particular single-cell libraries were prepared using the Nugen Ovation RNA-seq System v2 kit, rather than SMART-Seq v4. The expression of a selection of highly variable and cluster marker genes is depicted in *Figure 4—figure supplement 1*.

The online version of this article includes the following figure supplement(s) for figure 4:

**Figure supplement 1.** Expression patterns of a selection of highly variable and cluster marker genes that show anatomical bias.

**Figure supplement 2.** PAX5 and SATB2 are expressed predominately in rostral dorsomedial and dorsolateral *Pet1* neurons while NR2F2 is expressed predominately in caudal *Pet1* neurons.

dorsal and lateral portions of the DR, however, where there are far fewer intersectionally labeled neurons, we observed a small number of *Slc17a8-cre; Pet1-Flpe* intersectionally marked neurons that were negative for VGLUT3 antibody staining, suggesting transient expression of *Slc17a8* (and

Slc17a8-cre) by these cells at an earlier time in their developmental history (or low Slc17a8 expression sufficient to drive Cre expression, but not VGLUT3 immunodetection).

Transcripts for Npy2r, encoding the neuropeptide Y receptor Y2, are strongly enriched in clusters six, seven, and ten, with less consistent expression in clusters eleven, thirteen, and eight, and only sporadic expression elsewhere (Figure 3C). In mid-rostral portions of the DR, we found that Npy2r-cre; Pet1-Flpe intersectionally marked cell bodies show a largely midline bias, with a greater density of cells ventrally than dorsally, and the occasional labeled cell body appearing more laterally (Figure 3C2–C3). In more caudal extents of the DR, Npy2r-cre; Pet1-Flpe intersectionally marked cell bodies appear to be concentrated more medially (Figure 3C4–C6).

Transcripts for Crh, encoding corticotropin-releasing hormone, are most highly enriched in neurons comprising cluster nine and to a lesser extent cluster five, with sporadic expression in other clusters (Figure 3D). Crh-cre; Pet1-Flpe intersectionally labeled neurons do not show an obvious overall anatomical bias, distributing widely throughout the DR (Figure 3D1–D6). At the most rostral levels of the DR, they appear to be more consistently medially and ventrally localized (Figure 3D1–D2), but additionally appear in the dorsal and lateral DR at mid-rostral levels, and are preferentially localized off the midline more ventrally in these same sections (in regions sometimes referred to as the ventrolateral wings) (Figure 2D3–D4). At the most caudal levels they distribute dorsally and ventrally, with an apparent gap between these two domains (Figure 3D5–D6).

The most molecularly distinct Pet1 neuron subtype we identified, cluster twelve Met-Slc17a8-Tph2-Pet1 neurons, shows highly specific enrichment for a number of transcripts, including P2ry1, encoding purinergic receptor P2Y1, which is only sporadically expressed in other clusters (Figure 3E). P2ry1-cre; Pet1-Flpe intersectionally marked neurons likewise show a strikingly unique anatomical distribution from the other subgroups examined, being largely restricted to the caudal DR where they are densely clustered dorsally, just beneath the aqueduct (Figure 3E5–E6). This is consistent with previous characterizations of Met-expressing Pet1/5-HT neurons (Okaty et al., 2015; Wu and Levitt, 2013), as well as other more recent characterizations (Huang et al., 2019; Kast et al., 2017; Ren et al., 2019). Notably the distribution of P2ry1-cre; Pet1-Flpe intersectional neurons within the cDR is distinct from Npy2r-cre; Pet1-Flpe intersectional neurons, and only partially overlaps with where Crh-cre; Pet1-Flpe intersectional neurons are found, arguing for Pet1/5-HT neuron subtype diversity within the caudal DR, consistent with (Kast et al., 2017).

It should be noted that the precise anatomical boundaries of the caudal DR (cDR), also referred to as B6 (Dahlstroem and Fuxe, 1964; Jacobs and Azmitia, 1992), are variably described in the literature. Alonso and colleagues divide B6 into dorsal and ventral sub-compartments, referred to as r1DRd and r1DRv, respectively, where 'r1' designates the putative developmental domain of origin of Pet1 neurons residing in this DR subregion (i.e. originating from r1, as opposed to isthmus) (Alonso et al., 2013). r1DRv likely corresponds to what others have described as the caudal portion of the 'interfascicular' DR (DRI), a medioventral band of DR cells flanked on either side by the medial longitudinal fasciculi. 5-HT neurons of the caudal DRI merge with the more dorsal B6 DR sub-nucleus roughly at the level of the DR where dorsolateral 5-HT neurons become sparse (coronal sections 5 and 6 in Figure 3; Hale and Lowry, 2011; Jacobs and Azmitia, 1992). Depending on the plane and angle of sectioning these caudal DRI cells also appear to merge with MR 5-HT neurons more ventrally, and it has been proposed that caudal DRI cells may be more similar to MR 5-HT neurons developmentally, morphologically, and hodologically than to DR 5-HT neurons (Commons, 2015; Commons, 2016; Hale and Lowry, 2011; Jacobs and Azmitia, 1992). In the present study, our designation of cDR is inclusive of r1DRd/r1DRv/caudal DRI/B6, as indicated in Figure 4A. Moreover, we do not discount the possibility that this region as drawn partially overlaps with what Alonso and colleagues would call the most dorsal portion of the caudal median raphe (MnRc), as the boundary between the MnRc and r1DRv is poorly defined. Thus, the territory between the cluster of Met-Slc17a8-Tph2-Pet1 neurons beneath the aqueduct in the cDR and the MR is difficult to classify strictly based on cytoarchitecture, underscoring the importance of alternative classification schemes, such as offered by transcriptomics.

## Manual scRNA-seq of Pet1-Intersectionally defined neuron populations

Having mapped the spatial distributions of intersectionally labeled Pet1 neuron subgroups, next we wanted to explore the correspondence of molecular subtype identity with DR subregions more comprehensively. To do this, we microdissected subdomains of the DR in a subset of the intersectional

mouse lines just described, dissociated and sorted fluorescently labeled neurons, harvested mRNA from single cells, and prepared scRNA-seq libraries (n = 70 single-cell libraries in total) using the SMART-Seq v4 kit, followed by Illumina sequencing (*Figure 4A*). Specifically, we separately microdissected and manually sorted *Slc6a4-cre; Pet1-Flpe* intersectionally labeled neurons from the dorsolateral DR (dl or dlDR, n = 10 cells), dorsomedial DR (dm or dmDR, n = 9 cells) ventromedial DR (vm or vmDR, n = 8 cells), and caudal DR (cDR, n = 6 cells), as schematized in *Figure 4A*. Additionally, we separately microdissected and manually sorted *Npy2r-cre; Pet1-Flpe* intersectional neurons from the rostral (rDR, n = 9 cells) versus caudal (n = 10 cells) DR, *P2ry1-cre; Pet1-Flpe* intersectional neurons from the cDR (n = 10 cells), and *Crh-cre; Pet1-Flpe* intersectional neurons from the rostromedial DR (n = 8 cells). We then used the fourteen *Pet1* neuron subtype identities derived from our 10X scRNA-seq data as a reference to 'query' the corresponding identities of our manually sorted and transcriptomically profiled single cells (using the Seurat functions FindTransferAnchors and Transfer-Data as described in *Stuart et al., 2019*). A summary of this analysis is shown in the dot plot in *Figure 4B*. We found that the majority of *Slc6a4-cre; Pet1-Flpe* dlDR neurons mapped to cluster two, with a smaller percentage of single cells mapping to clusters one, three, and six. *Slc6a4-cre; Pet1-Flpe* dmDR neurons were split between clusters two and three, and to a lesser extent four, five, and nine. *Slc6a4-cre; Pet1-Flpe* vmDR neurons mostly corresponded to cluster ten, and were additionally mapped to clusters one, four, seven, and thirteen (note, this may suggest that some *Pet1* neurons expressing little or no *Slc6a4* nor *Tph2* in the adult may yet express the *Slc6a4-cre* transgene). Finally, *Slc6a4-cre; Pet1-Flpe* cDR neurons mapped exclusively to clusters eight and seven (note, cluster twelve neurons do not appear to be well marked by *Slc6a4-cre; Pet1-Flpe*; RC-FrePe EGFP expression – see *Figure 3A5* compared with *Figure 3E5* – perhaps reflecting the lower levels of *Slc6a4* transcripts detected in these neurons).

The majority of *Npy2r-cre; Pet1-Flpe* neurons in the rDR were found to correspond to cluster six, with additional mapping to clusters five, seven, ten, and thirteen (consistent with the expression profile of *Npy2r* transcripts in the 10X scRNA-seq data) whereas the majority of *Npy2r-cre; Pet1-Flpe* neurons from the cDR were found to correspond to cluster seven, with a smaller percentage corresponding to clusters ten and fourteen. *P2ry1-cre; Pet1-Flpe* cDR manually sorted and profiled neurons were mapped exclusively to cluster twelve as expected. *Crh-cre; Pet1-Flpe* profiled neurons were split across clusters in a manner consistent with sporadic *Crh* expression in our 10X scRNA-seq data, however, we found more cluster three than cluster nine *Crh-cre; Pet1-Flpe* neurons, perhaps reflecting that our sampling of this population was biased towards rostromedial DR (or a potential discrepancy between endogenous *Crh* expression and *Crh-cre* expression). Finally, we also included *Drd2-cre; Pet1-Flpe* intersectional scRNA-seq data (n = 17 cells) associated with a previous study from our lab (*Niederkofler et al., 2016*). *Drd2-cre; Pet1-Flpe* intersectional neurons show a largely dorsolateral and dorsomedial bias within the DR. The majority of these neurons map to clusters three and four, with a much smaller percentage mapping to clusters eight and eleven.

Thus combining intersectional genetics, histological analyses, and precisely targeted manual scRNA-seq we were able to infer the anatomical distributions of our fourteen clusters to varying degrees of specificity. Clusters one through six appear to be rostrally and dorsally biased, with cluster two showing a strong dorsolateral bias as well. Clusters seven, eight, and twelve appear to be caudally and medially biased, with cluster twelve showing a clear dorsal bias and clusters seven and eight showing more ventral bias based on *Figures 3* A5-6, C5-6 (though a nontrivial degree of intermixing of different genetically defined *Pet1* neuron subpopulations in the dorsal cDR is apparent from these images). *Pet1-Tph2*$^{low}$ neurons (comprising cluster thirteen and to a lesser extent cluster fourteen neurons) show a prominent enrichment in the dorsomedial and medial-rostral DR, though they are also scattered throughout the DR (but very rarely found dorsolaterally). The remaining clusters appear to be more ventromedially biased in the more rostral DR. Expression patterns of cluster marker genes showing strong anatomical biases in our manual scRNA-seq data are depicted in the dot plot in *Figure 4—figure supplement 1* in comparison with our 10X scRNA-seq data.

These inferred anatomical distributions of molecularly distinct *Pet1* neuron populations shed further light on the potential developmental significance of transcription factor expression patterns described above. As noted, *Pax5*, a gene associated with isthmic organizer activity during embryonic development (*Funahashi et al., 1999*; *Ye et al., 2001*), shows a complementary expression pattern to *Nr2f2*, which encodes a transcription factor that appears to be excluded from the isthmus, but is expressed in r1 and other rhombomeres during development, at least in zebrafish (*Love and Prince,*

*2012*). We further validated the anatomical expression profile of these genes, as well as *Satb2* (expressed by cluster two through four), at the level of protein expression by performing immunohistology in tissue sections prepared from *Slc17a8-cre; Pet1-Flpe; RC-FL-hM3Dq* mice (*Figure 4—figure supplement 2A–E*). Consistent with our anatomically-targeted, manual scRNA-seq data, PAX5 and SATB2 display a rostrodorsal bias in predominately non-*Slc17a8*-expressing DR *Pet1* neurons (*Figure 4—figure supplement 2B–C,E*), whereas NR2F2 has a ventromedial and caudal expression bias in predominately *Slc17a8*-expressing DR *Pet1* neurons (*Figure 4—figure supplement 2D,E*). Alonso and colleagues have proposed that cDR *Pet1* neurons are derived from r1 progenitors, whereas more rostral *Pet1* neurons are derived from isthmus (*Alonso et al., 2013*), however further fate-mapping experiments would be helpful to clarify isthmic versus r1-derived *Pet1* neuron populations (*Okaty et al., 2019*). Moreover, while rostral DR *Pet1* neurons may derive from isthmus and cDR *Pet1* neurons may derive from r1, our scRNA-seq data nonetheless show substantial *Pet1* neuron molecular heterogeneity within both DR domains, suggesting factors beyond isthmus and r1-lineage driving molecular diversity.

## cDR *P2ry1-cre; Pet1-Flpe* neurons display unique hodological and electrophysiological properties

Having established correlations between DR *Pet1* neuron molecular expression profiles and anatomical distribution of cell bodies, we next wanted to explore corresponding differences in other cellular phenotypes. We chose to focus on cluster twelve *Met-Slc17a8-Tph2 Pet1* neurons, captured intersectionally by *P2ry1-cre; Pet1-Flpe,* as they are the most distinct from other *Pet1* neurons molecularly. To determine if these neurons are likewise unique from other DR *Pet1* neurons with respect to other features we explored the hodological and electrophysiology properties of *P2ry1-cre; Pet1-Flpe* neurons using the intersectional expression of TdTomato (*GT(ROSA)26Sor^{tm65.1(CAG-tdTomato)Hze}*, referred to as RC-Ai65). The anatomical location of cell somata labeled in *P2ry1-cre; Pet1-Flpe; RC-Ai65* animals was similar to that found in the previously characterized *P2ry1-cre; Pet1-Flpe; RC-FrePe* mice, with a dense population of neurons directly under the aqueduct in the cDR. In addition, there were slightly higher numbers of intersectionally labeled cells in the rostral part of the dorsal raphe as well as scattered cells in the median raphe, consistent with the sporadic expression of *P2ry1* revealed by the present RNA-seq data and the scRNA-seq data of *Pet1* neurons from the MR (*Okaty et al., 2015*; *Ren et al., 2019*). Strikingly, most fibers from *P2ry1-cre; Pet1-Flpe; RC-Ai65* neurons were supra-ependymal and were found throughout the third, lateral, and fourth ventricles, a property previously attributed to 5-HT neurons within the cDR (*Kast et al., 2017*; *Mikkelsen et al., 1997*; *Tong et al., 2014*). Sparser fibers were found in regions such as the lateral hypothalamus, medial and lateral septum, hippocampus, olfactory bulb, lateral parabrachial nucleus, and the amygdala. To gain a better perspective of the extent of *P2ry1-cre; Pet1-Flpe; RC-Ai65* fibers in the lateral ventricle we stained for *P2ry1-cre; Pet1-Flpe; RC-Ai65* fibers on a flat mount of the lateral wall as previously described (*Mirzadeh et al., 2010*). *P2ry1-cre; Pet1-Flpe; RC-Ai65* fibers were found on all aspects of the wall except for the adhesion area, including regions that contain proliferating cells and migrating neuroblasts from the subventricular zone (*Mirzadeh et al., 2010*; *Figure 5*). Further, *P2ry1-cre; Pet1-Flpe; RC-Ai65* fibers were closely apposed to proliferating cells (Ki67+) and migrating neuroblasts (doublecortin, DCX+) within the subventricular zone (SVZ) and within the rostral migratory stream (RMS) (*Figure 5*). The proximity of *P2ry1-cre; Pet1-Flpe; RC-Ai65* fibers to adult neural stem cells suggests that they may constitute a serotonergic population of neurons that regulate SVZ proliferation, a process known to be regulated by 5-HT levels and that has previously been associated with the cDR (*Aghajanian and Gallager, 1975*; *Banasr et al., 2004*; *Brezun and Daszuta, 1999*; *Hitoshi et al., 2007*; *Kast et al., 2017*; *Lorez and Richards, 1982*; *Mirzadeh et al., 2010*; *Negoias et al., 2010*; *Siopi et al., 2016*; *Soumier et al., 2010*; *Tong et al., 2014*).

To determine if supra-ependymal projections are unique to *Pet1* neurons in the caudal dorsal raphe, we injected a retrograde AAV virus leading to expression of Cre under the synapsin promoter (pENN.AAV.hSyn.Cre.WPRE.hGH) unilaterally into the lateral ventricle of double transgenic *Pet1-Flpe; RC-FrePe* or *Pet1-Flpe; RC-Ai65* mice, where expression of both Cre and Flpe leads to cell labeling by EGFP or TdTomato respectively (*Figure 5—figure supplement 1A*). The predominant labeled population in both genotypes was in the cDR, just under the aqueduct, suggesting that *P2ry1-cre; Pet1-Flpe* neurons constitute the major supraependymal projecting group of *Pet1* neurons (*Figure 5—figure supplement 1B,C*). However, in agreement with other studies that have included

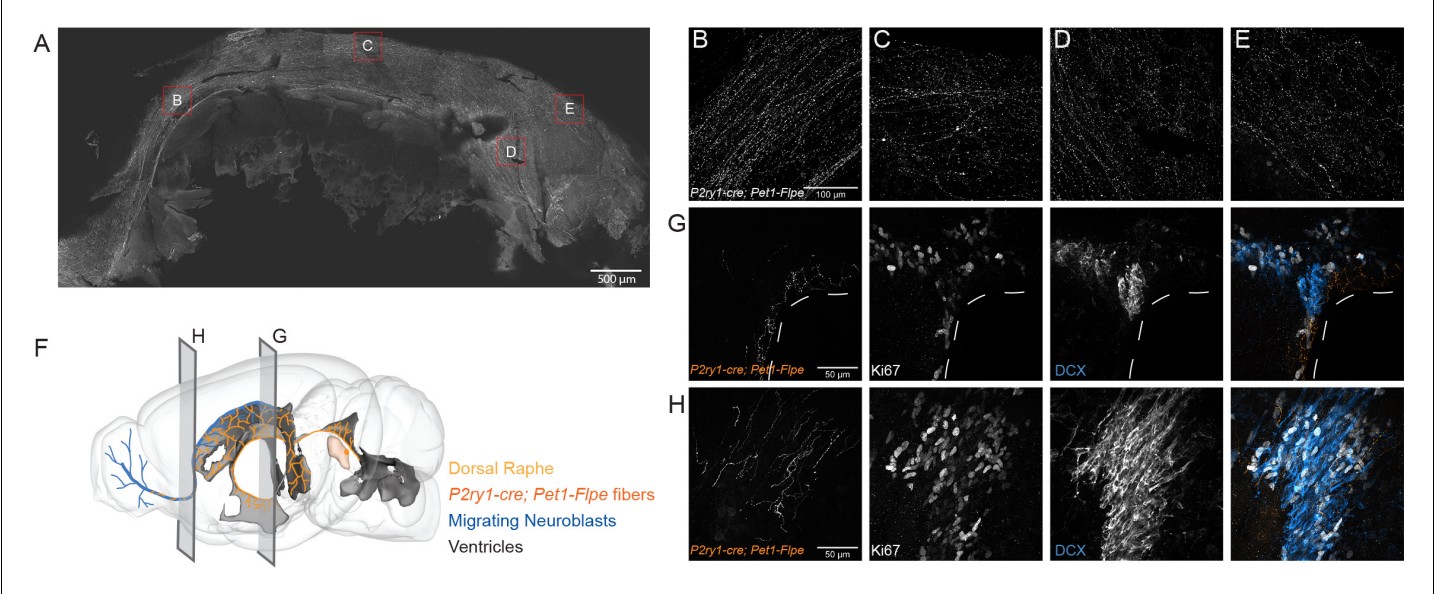

**Figure 5.** *P2ry1-cre; Pet1-Flpe* neurons project throughout the ventricles and their fibers are in close apposition to proliferating cells in the SVZ and RMS. (**A**) Flat mount of the lateral wall of the lateral ventricle of a *P2ry1-cre; Pet1-Flpe; RC-Ai65* animal, where *P2ry1-cre; Pet1-Flpe* fibers are in grey. Scale bar = 100 µm. (**B–E**) High magnification confocal images from regions of the lateral wall represented in red boxes in A. Scale bar (**B**) = 100 µm. (**F**) 3D brain schematic showing the *P2ry1-cre; Pet1-Flpe* cell bodies (dark orange) in the caudal part of the DR (light orange) and fibers (dark orange) projecting through the ventricles (grey) and along the migrating neuroblasts of the rostral migratory stream (RMS, blue). (**G–H**) Coronal confocal images depicting *P2ry1-cre; Pet1-Flpe* fibers (orange) from *P2ry1-cre; Pet1-Flpe; RC-Ai65* animals in the SVZ (**G**) and RMS (**H**). Proliferating cells labeled with Ki67 (grey) and migrating neuroblasts labeled with doublecortin (DCX, blue). Scale bar (**G, H**) = 50 µm.

The online version of this article includes the following figure supplement(s) for figure 5:

**Figure supplement 1.** The caudal dorsal raphe is the major *Pet1* neuron source of supra-ependymal fibers.

retrograde labeling via the lateral ventricle, some cell bodies were also found in the median raphe (*Kast et al., 2017*; *Tong et al., 2014*). Thus, supra-ependymal projections, while predominantly originating from the cDR, are not entirely unique to this region.

We next characterized electrophysiological properties of *P2ry1-cre; Pet1-Flpe; RC-Ai65* neurons in comparison with other more broadly defined *Pet1* neuron subpopulations using whole-cell patch clamp in acute slice preparations. As comparison groups we chose: (1) 'subtractive' *P2ry1-cre; Pet1-Flpe; RC-FL-hM3Dq* neurons in the cDR (i.e. cDR neurons with a history of *Pet1-Flpe* expression but not *P2ry1-cre* expression; we chose to use RC-FL-hM3Dq as opposed to RC-Ai65 or RC-FrePe because the subtractive population is identifiable in acute brain slices by EGFP fluorescence without the need for secondary staining), and (2) *Pet1* neurons from the more rostral and mostly dorsal DR using labeled intersectional expression of *Gad2^{tm2(cre)Zjh}* (referred to as *Gad2-cre*) with *Pet1-Flpe; RC-Ai65* (*Figure 6*). Recording from *P2ry1-cre; Pet1-Flpe; RC-FL-hM3Dq* subtractive cDR neurons allowed us to assess the degree to which electrophysiology may differ within a given DR subdomain depending on molecularly-defined neuron subtype, whereas *Gad2-cre; Pet1-Flpe; RC-Ai65* neuron recordings provided a comparison group that is both anatomically and molecularly distinct. As demonstrated in the frequency-current (F-I) curves in *Figure 6A*, we found that *P2ry1-cre; Pet1-Flpe; RC-Ai65* neurons have dramatically lower excitability than the two comparison populations, requiring substantially more injected current to reach action potential threshold, and showing a roughly three-fold lower maximum firing rate. Even within the regime of current injection that *P2ry1-cre; Pet1-Flpe; RC-Ai65* neurons are excitable, we found that they displayed very different spiking characteristics from other *Pet1* neuron groups (*Figure 6B,C*), specifically showing a longer latency to first action potential (AP, *Figures 6B* and *3*). Altogether, we observed four distinct firing types exemplified by the voltage traces displayed in *Figure 6B*: short-latency to first AP (regular spiking/non-adapting) (*Figures 6B* and *1*), mid-latency to first AP (*Figures 6B* and *2*), long-latency to first AP (*Figures 6B* and *3*), and short-latency to first AP with spike frequency adaptation (*Figures 6B* and *4*). The

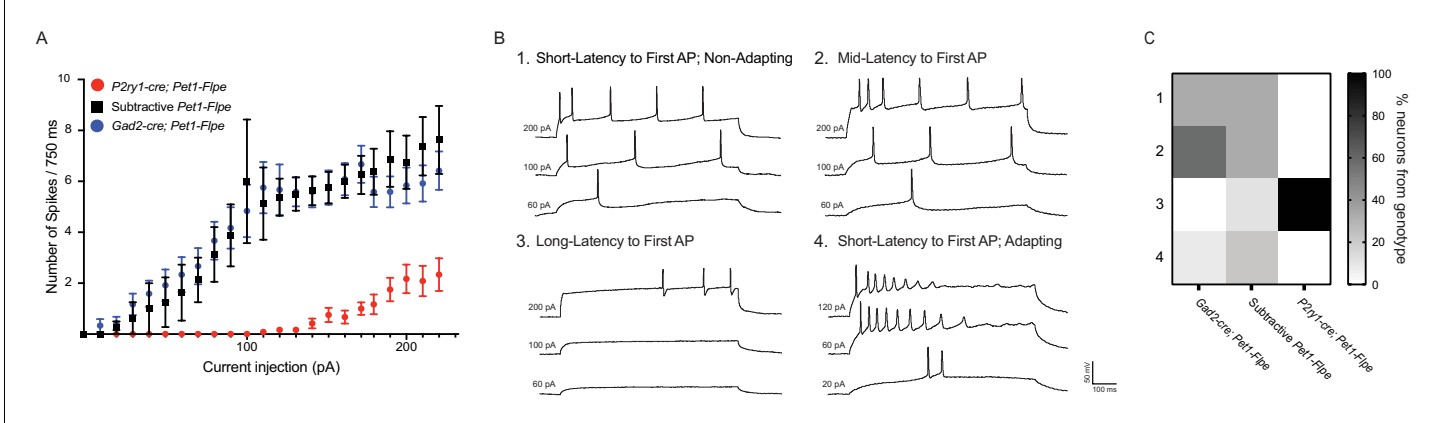

**Figure 6.** *P2ry1-cre; Pet1-Flpe* neurons have a distinct firing phenotype. (**A**) Frequency-Current (**F–I**) curves show *P2ry1-cre; Pet1-Flpe* neurons (tdTomato$^+$ *P2ry1-cre; Pet1-Flpe*; RC-Ai65, n = 12; three animals; red circles) are less excitable than nearby caudal dorsal raphe non-*P2ry1-cre; Pet1-Flpe* populations (EGFP$^+$ *P2ry1-cre; Pet1-Flpe*; RC-FL-hM3Dq, n = 8; three animals; black squares) or neurons from the dorsomedial and dorsolateral dorsal raphe *Gad2-cre; Pet1-Flpe* population (tdTomato$^+$ *Gad2-cre; Pet1-Flpe*; RC-Ai65, n = 12; two animals; blue circles) p<0.0001 Kruskal-Wallis test. (**B**) Example voltage traces from neuron patch-clamp recordings showing different firing types, specifically a neuron that started firing action potentials with (1) short latency (mean = 17.32 ms±6.61 at 200 pA), in response to 750 ms current pulses, (2) medium latency (mean = 64.18 ms±9.8 at 200 pA), (3) long latency (mean = 476.55 ms±223.64 at 200 pA), or (4) short latency (mean = 12.6 ms±5.9 at 200 pA) with spike-frequency adaptation. (**C**) Heat map shows the percentage of cells recorded from each genotype corresponding to each firing type, note all recorded *P2ry1-cre; Pet1-Flpe* neurons belong to type 3.

The online version of this article includes the following figure supplement(s) for figure 6:

**Figure supplement 1.** Key membrane properties distinguish serotonergic neuron firing types.

heatmap in *Figure 6C* shows the percentage of single-neuron recordings from each genotype that correspond to a given firing type. *Figure 6—figure supplement 1* displays differences in measured electrophysiological properties when cells are grouped by firing type, as opposed to genotype. All *P2ry1-cre; Pet1-Flpe; RC-Ai65* neurons recorded (twelve neurons from three animals) showed long latency to first AP, whereas only one out of nine subtractive neurons in the *P2ry1-cre; Pet1-Flpe; RC-FL-hM3Dq* cDR (from three animals) showed this phenotype and none of the *Gad2-cre; Pet1-Flpe; RC-Ai65* neurons (twelve neurons from two animals). These latter two groups of neurons showed greater heterogeneity with respect to firing characteristics, as might be expected given that labeled cells from both genotypes comprise multiple molecular subtypes identified by our scRNA-seq experiments. While the full extent of electrophysiological heterogeneity of these populations is likely under-sampled by the present dataset, the uniqueness of *P2ry1-cre; Pet1-Flpe; RC-Ai65* neurons nonetheless stands out.

## Comparison to other DR scRNA-seq datasets

Recent scRNA-seq studies of mouse DR cell types have been published (*Huang et al., 2019*; *Ren et al., 2019*), reporting using either the InDrops platform to profile dissociated DR neurons (*Huang et al., 2019*) or fluorescence-activated cell sorting to purify dissociated Cre-dependent tdTomato-expressing *Slc6a4-cre* neurons from mouse DR and MR, followed by SMART-Seq v2 library preparation and sequencing (*Ren et al., 2019*). Huang and colleagues identified six distinct *Pet1*-expressing DR neuron subtypes – five serotonergic and one glutamatergic – while Ren and colleagues identified seven *Pet1*-expressing serotonergic DR neuron subtypes (note they did not identify a glutamatergic *Tph2*$^{low}$ group, presumably because these neurons do not typically express *Slc6a4-cre*). To directly compare our subtype classifications, we used the fourteen *Pet1* neuron subtype identities derived from our 10X scRNA-seq data as a reference to query the corresponding identities of the Huang and Ren datasets (using the Seurat functions FindTransferAnchors and TransferData, as described above for comparison with our manual scRNA-seq data). The results of this analysis are shown in the dot plot in *Figure 7*. Some *Pet1* neuron subgroup classifications were highly consistent across studies. For example, one hundred percent of single neurons making up the

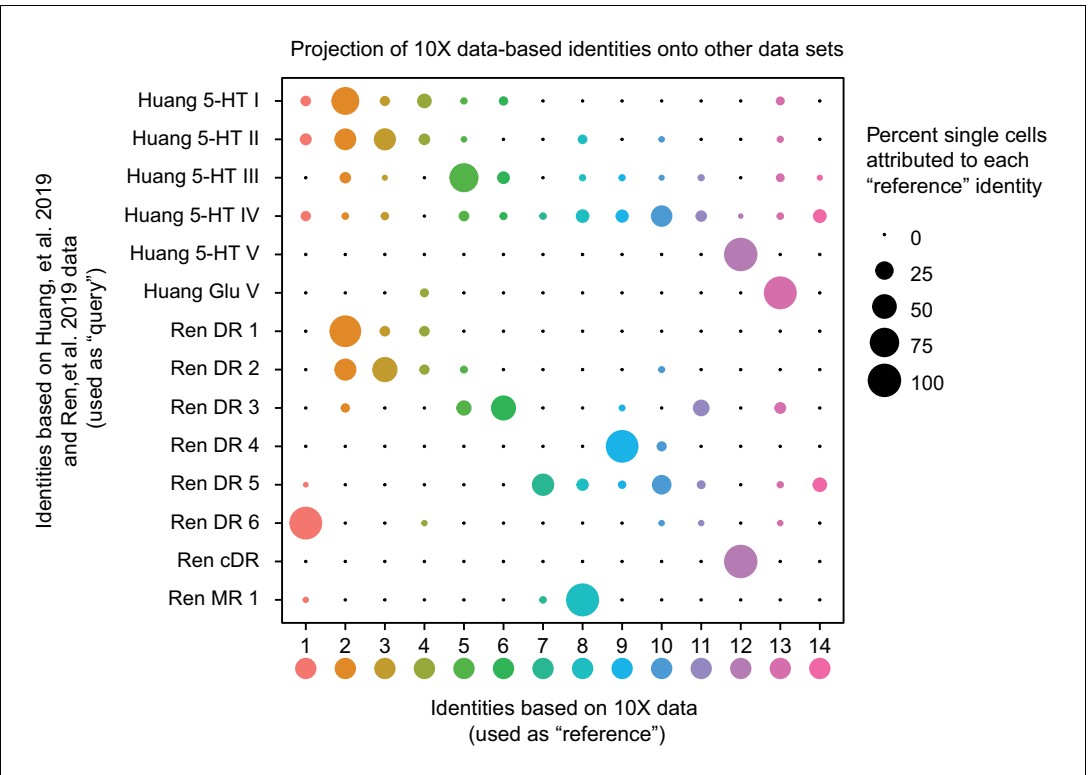

**Figure 7.** Correspondence of serotonin subtypes identified in previous studies (Y-axis) to the fourteen 10X scRNA-seq clusters identified in this study (X-axis). The size of the dot indicates the percentage of single cells from the original cluster that are attributed to a given reference cluster.

The online version of this article includes the following figure supplement(s) for figure 7:

**Figure supplement 1.** The number of cells sampled and the max UMIs per cell influence the number of clusters found in the dataset.

Huang 5-HT V and Ren cDR subgroups map to our cluster twelve *Pet1* neuron subgroup (*Met-Slc17a8-Tph2-Pet1* neurons, corresponding to neurons intersectionally captured by *P2ry1-cre; Pet1-Flpe* expression in the cDR shown in *Figure 3E*). Likewise, there is high correspondence between Huang 5-HT I, Ren DR 1, and our cluster two *Pet1*-neuron subgroup (dorsolateral DR *Gad2-Trh-Tph2-Pet1* neurons). Huang 5-HT II and Ren 2 subgroups are largely split between our cluster two and cluster three subgroups, with a smaller portion of each mapping to our cluster four subgroup (which also corresponds to a small subset of Huang 5-HT I and Ren DR 1 neurons). Huang 5-HT III and Ren DR 3 correspond to our cluster five and six subgroups, with a larger percentage of Huang 5-HT III neurons mapping to cluster five and a larger percentage of Ren DR 3 neurons mapping to cluster six.

In other cases, there is better correspondence between our identified *Pet1* neuron subgroups and one or the other study, likely due in part to technological differences between studies. For example, Huang Glu V corresponds well with our cluster thirteen *Pet1* neuron subgroup (*Slc17a8-Tph2^{low}-Pet1* neurons) but very few neurons profiled in the Ren study map to cluster thirteen. As mentioned above, the absence of a prominent glutamatergic *Tph2^{low}* group of neurons in the Ren study likely stems from the fact that the low level of *Slc6a4* transcription in these neurons does not reliably drive *Slc6a4-cre* transgene expression and thus reporter expression for their cell sorting. However, the fact that a small number of *Slc6a4-cre* expressing neurons from the Ren study do map to our cluster thirteen subgroup indicates that there may be exceptions (moreover, these cells may more specifically map to cluster thirteen neurons at the higher end of the distribution of *Slc6a4* and *Tph2* transcript levels *Figure 1—figure supplement 3*).

On the other hand, Ren DR 6 corresponds well with our cluster one subgroup (*Npas1/3-Tph2-Pet1* neurons) and Ren DR 4 corresponds well with our cluster nine subgroup (*Maf-Nos1-Tph2-Pet1*

neurons), but there is no such one-to-one correspondence between these groups and the neuron groups identified in the Huang study. Rather, cluster one and cluster nine neurons get 'pulled' from other groups identified by Huang. This likely reflects the different sensitivities of the various approaches. Specifically, our study and the Ren study achieved more than three-fold higher gene detection per single cell library on average than the Huang study, thus allowing for finer-scale molecular subgroup classification. However our higher single-cell sampling resolution – we profiled 2,350 DR *Pet1* neurons, whereas Huang and Ren profiled roughly 700 and 600 DR *Pet1* neurons, respectively – likely allowed us to resolve more subgroups. Huang 5-HT IV and Ren DR 5 show the greatest degree of dispersion into different clusters identified in our study. Ren DR 5 is split predominately between clusters seven, ten, and fourteen, while Huang 5-HT IV is split into clusters ten and fourteen, as well as across several other clusters. Lastly, we also found that Ren MR 1 (identified as being a median raphe 5-HT neuron subtype in that study) shows similarity to our cluster eight subgroup, which we have mapped to the cDR based on histology and our manual scRNA-seq data. As described above, the boundary between the cDR and the MR, specifically the portion of the MR attributed to r1-derived neurons (*Alonso et al., 2013*; *Okaty et al., 2015*), is poorly defined, thus Ren MR 1 and our cluster eight neurons may indeed partially overlap anatomically. Notably, some Huang 5-HT IV neurons (microdissected from what was considered the DR in that study) also map to our cluster eight subgroup.

Importantly, our higher number of identified clusters does not appear to stem from analytic differences between studies per se, given that we identified more clusters using a lower resolution parameter in our clustering analysis than the other two studies. Huang, et al. reported using a Seurat FindCluster resolution of 2.0, Ren, et al. used a resolution of 1.0, whereas the highest resolution we used was 0.9. In order to explicitly investigate how specific aspects of our experimental design may have led to identification of more DR *Pet1* neuron subgroups than the other two studies, we modeled how the number of cells sampled and the number of UMIs identified per cell influence the number of identified clusters in our dataset. Specifically, we randomly sub-sampled variable numbers of cells (twenty times per iteration) and re-clustered the resulting data in the same manner described for the full dataset (using a Seurat FindClusters resolution of 0.9), varying the number of sampled cells from 200 to 2,300 (*Figure 7—figure supplement 1A*). As expected, we found that increasing the number of cells increased the number of clusters. Identification of fourteen clusters did not occur until at least 1,700 cells were included in the analysis, and this number of clusters began to stabilize as 2,100 or more cells were included. Similarly, we randomly sub-sampled UMIs, varying the maximum number of UMIs per cell from 500 to 100,000, and repeated our clustering analysis twenty times per sub-sampled max UMI (*Figure 7—figure supplement 1B*). In this case we found a much steeper relationship between max UMIs and number of clusters identified, with fourteen clusters being identified with as few as 4,500 max UMIs per cell, and completely stabilizing at roughly 60,000 max UMIs. With respect to our ability to resolve fine-scale DR *Pet1* neuron subgroup structure, these results indicate that while both variables are important, the number of cells sampled was more limiting than the number of UMIs in our dataset; that is, we could have uncovered a similar degree of overall cellular diversity (fourteen subtypes) with less 'complex' libraries (e.g. from more shallow sequencing), however we needed nearly 90% of the cells we sampled to consistently uncover fourteen molecular subgroups.

The results of these analyses shed light on the most likely reasons why we were able to achieve more fine-grained classification of DR *Pet1* neuron subtypes than the other two studies. For example, the Ren study had a similar degree of library complexity to ours (slightly higher, in fact), however as noted above they profiled fewer cells – 567 to our 2,350 cells (~2,200 excluding *Tph2-low* cells). When we sub-sample our data to a similar number of cells, we find between six and nine clusters, and similarly, Ren, et al. reported seven DR 5-HT neuron clusters. When we simultaneously sub-sample both the number of cells profiled and the maximum number of UMIs detected per single cell to levels similar to the Huang, et al. study (750 cells and 2,500 max UMIs, *Figure 7—figure supplement 1C*), we uncover between four and seven subgroup clusters – Huang, et al. found six. Thus, all other methodological sources of variation between studies aside, these two parameters plausibly explain differences in the degree of diversity uncovered across studies.

## Discussion

The dorsal raphe nucleus is likely one of the most extensively connected hubs in the mammalian brain. Efferent DR fibers, predominantly serotonergic (but also glutamatergic and GABAergic), collectively innervate much of the forebrain and midbrain, as well as some hindbrain nuclei (*Azmitia and Segal, 1978*; *Bang and Commons, 2012*; *Bang et al., 2012*; *Beaudet and Descarries, 1976*; *Fernandez et al., 2016*; *Gagnon and Parent, 2014*; *Hale and Lowry, 2011*; *Kast et al., 2017*; *Kosofsky and Molliver, 1987*; *Lidov et al., 1980*; *Lidov and Molliver, 1982*; *Maddaloni et al., 2017*; *McDevitt et al., 2014*; *Molliver, 1987*; *Muzerelle et al., 2016*; *O'Hearn and Molliver, 1984*; *Prouty et al., 2017*; *Ren et al., 2018*; *Steinbusch, 1981*; *Steinbusch et al., 1980*; *Vasudeva et al., 2011*; *Vertes, 1991*; *Vertes and Kocsis, 1994*), and DR afferents have been identified from as many as eighty distinct anatomical brain regions, including other brainstem raphe nuclei (*Celada et al., 2001*; *Commons, 2015*; *Gonçalves et al., 2009*; *Levine and Jacobs, 1992*; *Mosko et al., 1977*; *Ogawa et al., 2014*; *Peyron et al., 1997*; *Peyron et al., 2018*; *Pollak Dorocic et al., 2014*; *Weissbourd et al., 2014*). As such, the DR is hodologically poised to send and receive signals related to a wide range of sensory, motor, affective, and cognitive processes. Indeed, DR neuropathology is associated with several human disorders (or disease models thereof) with broad symptomatology, such as major depressive disorder, autism, and Alzheimer's disease (*Chen et al., 2000*; *Dengler-Crish et al., 2017*; *Autism Sequencing Consortium et al., 2019*; *Ellegood et al., 2015*; *Guo and Commons, 2017*; *Ji et al., 2020*; *Luo et al., 2017*; *Michelsen et al., 2008*; *Miyazaki et al., 2005*; *Šimić et al., 2017*; *Vakalopoulos, 2017*; *Wang et al., 2018*; *Zweig et al., 1988*). Outside of DR-specialist research, the DR has often been viewed by the wider neuroscience community as a 'black box' source of a single neurochemical, namely 5-HT. Accordingly, development of therapeutics for associated disorders has largely focused on modulating overall serotonergic tone. However, DR-focused research over several decades has revealed layers of functional complexity and compositional heterogeneity warranting a more nuanced view (reviewed in *Abrams et al., 2004*; *Andrade and Haj-Dahmane, 2013*; *Gaspar and Lillesaar, 2012*; *Hale and Lowry, 2011*; *Michelsen et al., 2007*; *Okaty et al., 2019*; *Vasudeva et al., 2011*). While these studies have reached into the black box of the DR and described a variety of features at different levels of observation, integration across levels to arrive at principles of DR organization has proved challenging. Elucidating how molecular, neurochemical, anatomical, hodological, electrophysiological, and functional descriptions of the DR overlap is essential to understanding the structure-function relationship of the DR and other raphe nuclei (*Brust et al., 2014*; *Fernandez et al., 2016*; *Huang et al., 2019*; *Kast et al., 2017*; *Niederkofler et al., 2016*; *Okaty et al., 2015*; *Prouty et al., 2017*; *Ren et al., 2018*; *Ren et al., 2019*), and will likely facilitate improved therapies for human disorders. Here we have focused on one broadly defined subgroup of DR cells – neurons that express the gene *Pet1/Fev* – and applied scRNA-seq, iterative intersectional genetics, histology, and slice electrophysiology to provide a transcriptomic and anatomic atlas of mouse DR *Pet1* neurons with examples of links between molecular, neurochemical, anatomical, hodological, and electrophysiological levels of description. We identify as many as fourteen distinct molecularly defined subtypes of *Pet1* neurons that show biased cell body distributions in DR subregions. We further characterize projections and electrophysiology of the most molecularly unique DR *Pet1* neuron subtype – *Met-Slc17a8-Tph2-Pet1* cDR neurons (cluster twelve), genetically accessed by intersectional *P2ry1-cre; Pet1-Flpe* expression. The present study complements other recent characterizations of DR cell types (*Huang et al., 2019*; *Ren et al., 2019*), increasing the sampling resolution of *Pet1* neurons in particular through our experimental approach to achieve fine-scale identification of *Pet1* neuron subtypes.

### Molecular and anatomic organization of *Pet1* neuron subtypes

Our data and analysis highlight the hierarchical organization of DR *Pet1* neurons molecularly and anatomically, allowing for identification of features that organize *Pet1* neurons at different levels of granularity (*Figure 8*, *Figure 8—source data 1*). Neurochemistry has long served as a principal phenotypic axis for classifying neurons, and concordantly we found that distributions of transcripts associated with distinct neurotransmitters correspond with broad subgroup divisions. The majority of *Pet1* neurons (clusters one through twelve) express high levels of *Tph2* mRNA, encoding tryptophan hydroxylase two, the rate-limiting biosynthetic enzyme for 5-HT, as well as several other genes

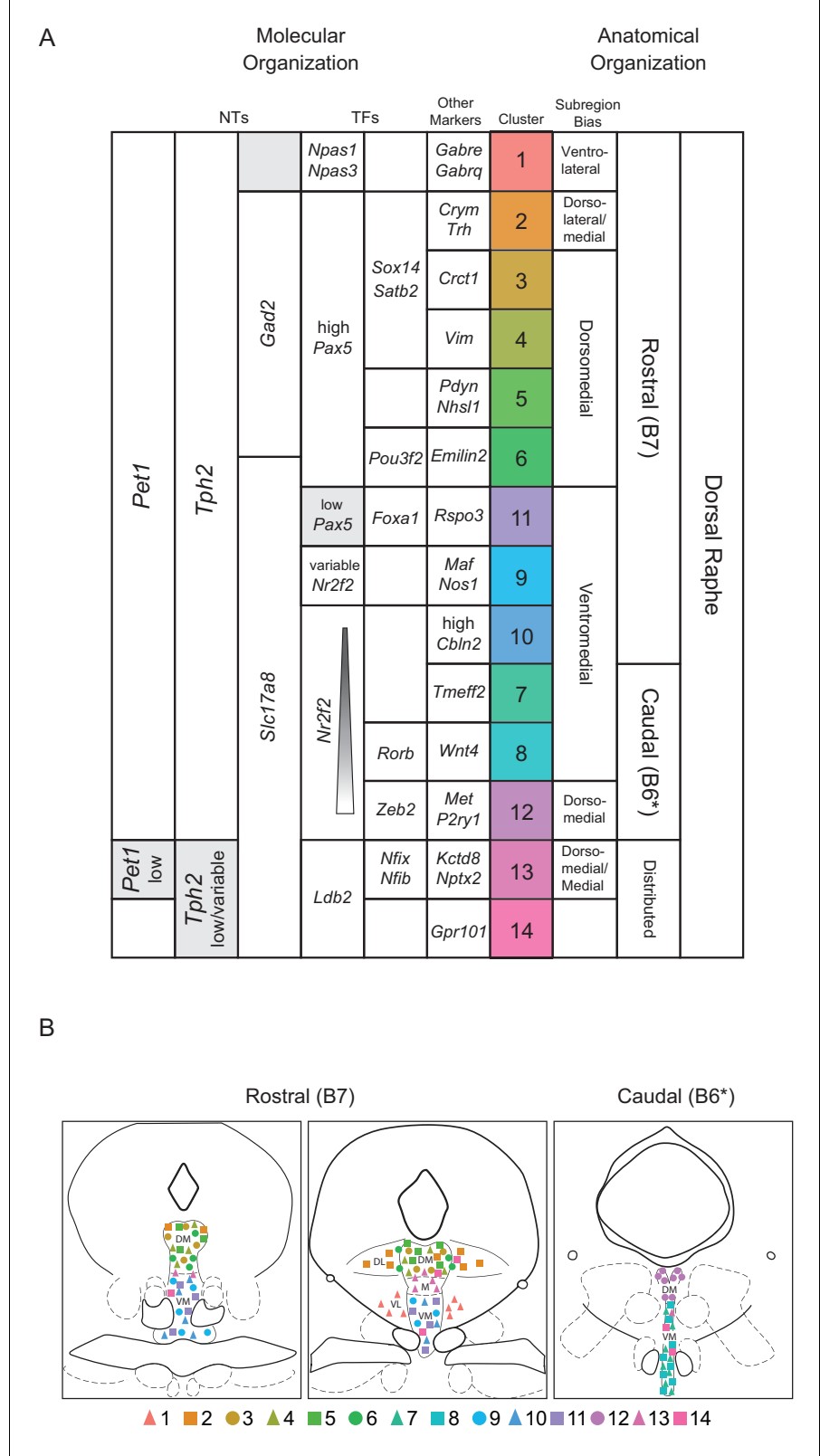

**Figure 8.** Fourteen *Pet1* subtypes in the DR can be defined by the combinatorial expression of transcription factors and other markers and have distinct anatomical organization. (**A**) Molecular markers (neurotransmitters (NTs), transcription factors (TFs), and other markers) on the left half of the table, with increasing specificity from left to right, that combinatorically define each identified *Pet1* subtype (colored column). Anatomical biases of each

*Figure 8 continued on next page*

*Figure 8 continued*

cluster are described on the right, with increasing specificity from right to left. *Figure 8—source data 1* outlines the combination of information used to inform the proposed anatomical bias. Note, cluster numbers have been re-ordered to highlight anatomical groupings. (**B**) Schematic depicting the anatomical distribution of each subtype based on the bias indicated in (**A**). B7 and B6 refer to the original Dahlström and Fuxe nomenclature for describing distinct anatomical clusters of 5-HT neurons. The asterisks after B6 in A and B are to indicate that some authors only consider B6 to encompass the dorsal part of what we refer to as the caudal DR.

The online version of this article includes the following source data for figure 8:

**Source data 1.** Anatomical bias of *Pet1* DR subtypes can be inferred by a combination of histology, single cell RNAseq, data from previously published papers, and Allen Mouse Brain atlas RNA in situ hybridization data.

indicative of a serotonergic phenotype, such as *Slc6a4* (Sert), *Slc18a2* (Vmat2), and *Maob*. However, we also identified two subgroups of *Pet1* neurons with 5-HT marker gene profiles that differ from the majority (clusters thirteen and fourteen). One subgroup (cluster thirteen) expresses very low transcript levels of 5-HT neuron marker genes, is mostly negative for TPH2 immunolabeling, and shows a biased cell body distribution in the rostromedial DR, as well as distributing sporadically throughout. The other subgroup (cluster fourteen) exhibits a much broader distribution of transcript levels for 5-HT marker genes than other groups. The functional significance of this variable expression can only be hypothesized at present; we speculate that it may reflect a capacity for neurotransmitter plasticity – that is experience-dependent induction or up-regulation of 5-HT phenotype, as hinted at by a recent study (*Prakash et al., 2020*). If this is the case, cluster fourteen neurons may be partially in transition, for example, from a predominately glutamatergic phenotype to a 5-HT phenotype or to a glutamate-5-HT co-transmitter phenotype. Both cluster thirteen and fourteen *Pet1* neuron subgroups express *Slc17a8* (alias Vglut3) transcripts, suggestive of a capacity for synaptic glutamate packaging and release, and show shared enrichment for several transcripts, including *Ldb2*, encoding the transcription factor LIM domain binding 2, and *Cnr1*, encoding cannabinoid receptor 1. Cluster fourteen is also uniquely distinguished by enrichment for *Gpr101* transcripts, encoding an orphan G protein-coupled receptor.

Among *Pet1* neurons expressing high levels of *Tph2* and other 5-HT gene markers, expression of genes related to GABA synthesis (*Gad1* and *Gad2*) or glutamatergic synaptic vesicle packaging (*Slc17a8*) correlate with major molecular and anatomical subdivisions (evident in the dendrogram in *Figure 1B*, the UMAP plot in *Figure 1C*, and the histological image series in *Figure 3* B1–6 and *Figure 3—figure supplement 1D–J*). We found that the cell bodies of non-*Slc17a8*, largely *Gad2*-expressing *Pet1* neurons are preferentially distributed in the dorsal and lateral subregions of the rostral DR, and become exclusively lateral and ultimately absent at more caudal extents of the DR. Conversely, *Slc17a8-Tph2-Pet1* neuron bodies show a ventromedial bias rostrally and predominate the entire cDR. *Gad2-Tph2-Pet1* neurons and *Slc17a8-Tph2-Pet1* neurons show differential enrichment of hundreds of transcripts, including *Pax5* and *Nr2f2*. Both genes encode transcription factors, the expression of which we examined through immunohistology and found a similar distribution of cell body staining as revealed by intersectional genetic labeling of *Slc17a8-cre; Pet1-Flpe* neurons – *Pax5* expression overlaps predominately with the non-*Slc17a8*-expressing population, whereas *Nr2f2* overlaps with the *Slc17a8*-expressing population (*Figure 4—figure supplement 2*). We also found one *Pet1* neuron subgroup (cluster six) that expresses *Gad1*, *Gad2*, and 'intermediate' levels of *Slc17a8* transcripts (relative to other *Slc17a8+* clusters). These neurons correspond with the rostral population of neurons labeled by intersectional *Npy2r-cre; Pet1-Flpe* expression (*Figure 3C*), which we characterized by manual scRNA-seq (*Figure 4*).

Altogether we identified five *Pet1* neuron subgroups that express *Gad1* or *Gad2* transcripts (clusters two through six), and found *Gad2* to be expressed more consistently and at higher levels than *Gad1* (with the exception of cluster six). *Gad1* and *Gad2* encode two distinct isoforms of glutamate decarboxylase, referred to as GAD67 and GAD65, respectively. In many neuron types, these proteins are often co-expressed, but localize to different subcellular compartments and differ in their interaction with the co-factor pyridoxol phosphate (*Chen et al., 2003*; *Erlander et al., 1991*; *Soghomonian and Martin, 1998*). GAD65 (encoded by *Gad2*) is typically found in axon terminals where it is thought to play a role in GABA synthesis specifically for synaptic vesicular release, whereas GAD67 is typically localized to the soma and may be more involved with non-vesicular

GABA release. While we did not reliably detect transcripts for the vesicular GABA transporter (*Slc32a1*), we did detect expression of transcripts encoding VMAT2 which has been shown to package GABA into synaptic vesicles in dopaminergic neurons, allowing for monoaminergic-GABAergic co-transmission (*Tritsch et al., 2012*). Thus it is plausible that *Gad2-Tph2-Pet1* neurons may likewise co-release GABA, though it has yet to be reported in the literature.

Beyond classic neurotransmitters, we also found enrichment of various peptide hormone transmitters in different *Gad2-Tph2-Pet1* neuron subgroups. Cluster two shows enrichment for thyrotropin-releasing hormone transcripts (*Trh*), and clusters five and six show enrichment for prodynorphin (*Pdyn*) (as does cluster fourteen). As can be seen in the dendrogram in *Figure 1B* and UMAP plot in *Figure 1C*, there appears to be a major division between *Gad2-Tph2-Pet1* clusters two through four and clusters five and six, with clusters five and six also sharing many molecular similarities with *Slc17a8-Tph2-Pet1* subgroups. This may in part reflect differential expression of transcription factors that regulate divergent gene 'modules'. While all *Gad2-Tph2-Pet1* neurons express *Pax5*, clusters two through four also express *Sox14* and *Satb2*. Cluster six, on the other hand, shows enrichment for several transcription factor genes that are also enriched (or trend towards enrichment) in clusters thirteen and fourteen, such as *Pou3f2*, *Bcl11a*, and *Id2*. These molecularly distinct *Gad2-Tph2-Pet1* subgroups also show differences in anatomy. Based on manual scRNA-seq of sub-anatomically targeted *Pet1* neurons, we found that cluster two *Gad2-Trh-Tph2-Pet1* neuron cell bodies are found predominately in the dorsolateral DR, as well as the dorsomedial DR, whereas clusters three, four, and five appear to be more dorsomedially biased, consistent with recent reports (*Huang et al., 2019*; *Ren et al., 2019*). Cluster six neurons, as captured by *Npy2r-cre; Pet1-Flpe* expression, show a more diffuse distribution in the rostral DR, illustrating that not all *Pet1* neuron subtypes, as defined transcriptomically, correspond with clear-cut anatomical patterns. Indeed, while there are major differences between predominately dorsal versus ventral or rostral versus caudal DR *Pet1* neuron subtypes, different subtypes nonetheless intermix within these domains, emphasizing the importance of molecular-genetic targeting of *Pet1* neuron subtypes to gain specificity for functional characterization (a point also made by *Huang et al., 2019* and *Okaty et al., 2019*).

*Slc17a8-Tph2-Pet1* neuron subgroups, as noted above, are found more ventrally than *Gad2-Tph2-Pet1* neurons in the rostral DR, and are the dominant neurotransmitter phenotype in the caudal DR (as inferred by transcript expression and VGLUT3 and TPH2 immunostaining). We found two *Slc17a8-Tph2-Pet1* neuron subtypes with cell bodies biased towards the rostral DR (clusters nine and ten) and three subtypes (clusters seven, eight, and twelve) biased towards the more caudal DR (as delineated in *Figure 4*, and see the discussion of varying nomenclature around the cDR in the Results section *Histology of Pet1-Intersectionally Defined Neuron Populations* above). Cluster twelve *Pet1* neurons, also marked by expression of the gene *Met*, we found to be the most different from all other *Pet1* neurons, both in terms of the number of differentially expressed genes and the magnitudes of enrichment/depletion compared to other *Pet1* neuron subtypes (as also observed by *Huang et al., 2019*; *Ren et al., 2019*). We show by histology of genetically marked neurons (intersectional *P2ry1-cre; Pet1-Flpe* expression) that the cell bodies of these *Met-Slc17a8-Tph2-Pet1* neurons are clustered beneath the aqueduct in the caudal DR (*Figure 3E*) and send extensive axonal projections throughout the ventricles (*Figure 5*). Based on our retrograde tracing experiments and other studies (*Kast et al., 2017*; *Tong et al., 2014*), it is likely that these neurons constitute the major source of 5-HT innervation to the ventricles. Furthermore, our demonstration that *P2ry1-cre; Pet1-Flpe* fibers are closely apposed with proliferating and migrating cells in the SVZ and rostral migratory stream (*Figure 5*), supports a proposed role for these neurons in regulating adult neural stem cell proliferation in the SVZ (*Tong et al., 2014*). Cluster twelve neuron transcript enrichment for several GPCRs implicated in modulation of adult neurogenesis, such as *P2ry1* (*Lin et al., 2007*), *Gipr* (found to be enriched in our *P2ry1-cre; Pet1-Flpe* manual scRNA-seq data) (*Nyberg, 2005*), *S1pr3* (*Alfonso et al., 2015*; *Ye et al., 2016*), and *Oprm1* (*Harburg et al., 2007*) lends further support to this hypothesis. Now, with intersectional access to this population of cDR 5-HT neurons provided by *P2ry1-cre* with *Pet1-Flpe,* the function of *Met-Slc17a8-Tph2-Pet1* cDR neurons in regulating SVZ proliferation can be tested directly in a cell type-specific manner using dual Cre- and Flpe-responsive chemo- or optogenetic approaches (*Brust et al., 2014*; *Hennessy et al., 2017*; *Kim et al., 2009*; *Madisen et al., 2015*; *Niederkofler et al., 2016*; *Okaty et al., 2015*; *Ray et al., 2011*; *Sciolino et al., 2016*; *Teissier et al., 2015*).

## DR *Pet1* neuron subtypes have distinct electrophysiological properties

To further characterize correspondence of molecular identities with other cell phenotypes we performed whole-cell electrophysiological recordings in acute slices prepared from mice in which different *Pet1* neuron subsets were genetically labeled. We found that 5-HT neurons with different molecular identities also exhibit distinct electrophysiological properties likely to impact their circuit function. While we did not comprehensively sample all molecularly defined subtypes, our survey of cDR *Pet1* neurons and rostral dorsal raphe *Gad2-cre; Pet1-Flpe* neurons provides evidence for at least four distinct electrophysiological types based on four key properties: (1) rheobase (also known as current threshold), which reflects a neuron's sensitivity to input, (2) delay to first spike, which reflects the degree to which a neuron is able to activate phasically in response to input, (3) spike-frequency adaptation, which reflects the degree to which a neuron is able to continuously signal ongoing input, and (4) maximum firing rate, which determines the dynamic range of neuron responsiveness to graded inputs. As with molecular differences, cluster twelve *Met-Slc17a8-Tph2-Pet1* cDR neurons (*P2ry1-cre; Pet1-Flpe* intersectional expression) showed profound differences from other subtypes, including other cDR *Pet1* neurons. *P2ry1-cre; Pet1-Flpe* neurons consistently displayed a long latency to first action potential, required substantially more input to reach action potential threshold, and had a lower maximum firing rate (*Figure 6* and *Figure 6—figure supplement 1*). These differences, together with differential transcript expression of several GPCRs, suggest that *Met-Slc17a8-Tph2-Pet1* cDR neurons respond in a different way and to very different stimuli than other DR *Pet1* neuron types. For example, low excitability and long-latency to spike suggest that these neurons may only be recruited by very strong stimuli at relatively slower timescales than other *Pet1* neurons (to the extent that properties recorded in slice reflect in vivo properties). Notably, 5-HT neurons with this electrophysiological profile have not yet been reported in the literature. However, the two firing types that we have defined as 'Short-Latency to First AP; Non-Adapting' and 'Mid-Latency to First AP' (*Figure 6B*, 1 and 2) correspond well to those described by *Fernandez et al., 2016* in groups of *Pet1*-EGFP serotonergic neurons projecting to the mPFC and the BLA, respectively. Differential expression of ion channels and receptors identified here suggest molecular substrates of these different electrophysiological properties.

## Technical aspects of our study allow for high-resolution transcriptome characterization of *Pet1* neurons

Due to the high-dimensional 'richness' of transcriptomic data, together with the capacity to propose explanations of cellular phenotypes in terms of molecular mechanisms – RNA-seq dissection of neural circuits has gained traction as a way to define and enumerate cell types in the brain (and other tissues). Single-cell RNA-sequencing, in particular, has become an indispensable approach, with different methods achieving different resolution of underlying cellular diversity (*Bakken et al., 2018*; *Campbell et al., 2017*; *Hodge et al., 2019*; *Huang et al., 2019*; *Lovatt et al., 2014*; *Macosko et al., 2015*; *Okaty et al., 2015*; *Poulin et al., 2016*; *Ren et al., 2019*; *Rosenberg et al., 2018*; *Saunders et al., 2018*; *Spaethling et al., 2014*; *Tasic, 2018*; *Tasic et al., 2016*; *Tasic et al., 2018*; *Usoskin et al., 2015*; *Zeisel et al., 2018*; *Zeisel et al., 2015*). Droplet-based scRNA-seq approaches (without cell-type-specific purification) allow for unbiased classification of major cell types residing in a particular microdissected tissue region of interest, however lower abundance cell types, such as DR *Pet1* neurons profiled in the present study, are often insufficiently sampled to achieve high resolution of subtype molecular diversity. Moreover, different reaction chemistries employed in different droplet-based scRNA-seq approaches can lead to different gene detection sensitivity. Low cellular abundance compounded with low gene detection can greatly limit the power of a study to reveal fine-scale variation in molecular phenotypes that may be important for identifying neuronal subtypes and subtype 'states' (e.g. adaptive or pathological transcriptional variation). Where cell type-specific markers are available, cell sorting prior to scRNA-seq library preparation can greatly enhance the resolution of cellular diversity for less abundant cell classes. While manual sorting approaches combined with RNA-seq library preparation optimized for low amounts of input RNA achieve high single-cell gene detection and allow for sampling genetically and anatomically-defined neuron populations (*Niederkofler et al., 2016*; *Okaty et al., 2015*), they are often limited in the number of cells profiled, and therefore may lack sufficient throughput to fully characterize subtype diversity. On the other hand, automated sorting approaches achieve greater throughput but

are less well suited to collecting low abundance cell types, such as defined by fine-scale anatomy or highly restricted marker gene expression. Our particular experimental approach to characterizing DR *Pet1*-lineage neuron diversity in the present study was informed by all of the above concerns. By combining intersectional genetic labeling of DR *Pet1* neurons with both high-throughput (microfluidic On-chip Sort) and targeted low-throughput (manual) sorting approaches, followed by high-sensitivity RNA-seq library preparation protocols (10X Genomics Chromium Single Cell 3' v3 and SMART-Seq v4 kits, respectively) we leveraged the strengths of multiple approaches to achieve high-resolution transcriptomic profiling of DR *Pet1* neurons.

## Resource value of DR Pet1 neuron scRNA-seq data

While we have highlighted many salient experimental findings in the present report, the data no doubt have more to reveal, and we thus offer this dataset as a resource to be mined by the larger community. Towards this end, we have created an interactive web application allowing users to directly explore our scRNA-seq dataset (https://dymeckilab.hms.harvard.edu/RNAseq_database). With the web app, users can plot the expression of a gene or several genes of interest, and perform differential expression analysis. Newly identified *Pet1* neuron subtype marker genes may guide development of new recombinase driver lines allowing for subtype-specific genetic access and functional manipulation, and may potentially shape approaches for developing more targeted therapeutics. Moreover, we hope this work, together with other recent studies (*Huang et al., 2019*; *Ren et al., 2019*), may lead to the development of a standardized DR *Pet1* neuron subtype nomenclature that allows for consolidation of experimental results across different labs and different data modalities.

# Materials and methods

**Key resources table**

| Reagent type (species) or resource | Designation | Source or reference | Identifiers | Additional information |
|---|---|---|---|---|
| Strain, strain background (*Mus musculus*) | C57BL/6J | The Jackson Laboratory | RRID:IMSR_JAX:000664 | |
| Genetic reagent (*Mus musculus*) | Tg(Fev-flpe)1Dym Referred to as *Pet1-Flpe* | PMID:18344997 | RRID:MGI:5004974 | |
| Genetic reagent (*Mus musculus*) | En1<sup>tm2(cre)Wrst</sup> Referred to as *En1-cre* | PMID:10837030 | RRID:IMSR_JAX:007916 | |
| Genetic reagent (*Mus musculus*) | Tg(Slc6a4-cre)ET33Gsat Referred to as *Slc6a4-cre* | PMID:17855595 | RRID:MGI:3836639 | |
| Genetic reagent (*Mus musculus*) | Npy2r<sup>tm1.1(cre)Lbrl</sup> Referred to as *Npy2r-cre* | PMID:25892222 | RRID:IMSR_JAX:029285 | Lab of Steve Liberles |
| Genetic reagent (*Mus musculus*) | P2ry1<sup>tm1.1(cre)Lbrl</sup> Referred to as *P2ry1-cre* | PMID:25892222 | RRID:IMSR_JAX:029284 | Lab of Steve Liberles |
| Genetic reagent (*Mus musculus*) | Tg(Crh-cre) KN282Gsat/Mmucd Referred to as *Crh-cre* | | RRID:MMRRC_030850-UCD | |
| Genetic reagent (*Mus musculus*) | Gad2<sup>tm2(cre)Zjh</sup> Referred to as *Gad2-cre* | PMID:21943598 | RRID:IMSR_JAX:028867 | |
| Genetic reagent (*Mus musculus*) | Slc17a8<sup>tm1.1(cre)Hz</sup> Referred to as *Slc17a8- cre* | MGI: J:146821 | RRID:IMSR_JAX:028534 | |
| Genetic reagent (*Mus musculus*) | Gt(ROSA)26Sor<sup>tm8(CAG-mCherry,-EGFP)Dym</sup> Referred to as RC-FrePe | PMID:22151329 | RRID:IMSR_JAX:029486 | |
| Genetic reagent (*Mus musculus*) | Gt(ROSA)26Sor<sup>tm65.1(CAG-tdTomato)Hze</sup> Referred to as RC-Ai65 | PMID:25741722 | RRID:IMSR_JAX:021875 | |

*Continued on next page*

Continued

| Reagent type (species) or resource | Designation | Source or reference | Identifiers | Additional information |
|---|---|---|---|---|
| Genetic reagent (*Mus musculus*) | GT(ROSA)26 Sor^tm3.2(Cag-EGFP,CHRM3*/mCherry/Htr2a)Pjen Referred to as RC-FL-hM3Dq | PMID:27264177 | RRID:IMSR_JAX:026942 | |
| Recombinant DNA reagent | pENN.AAV.hSyn.Cre.WPRE.hGH (AAVrg Viral prep) | Addgene | Cat# 10553-AAVrg RRID:Addgene_105553 | |
| Antibody | anti-GFP (chicken polyclonal) | Aves Labs | Aves Labs Cat# GFP-1020, RRID:AB_10000240 | IHC (1:3000) |
| Antibody | anti-DsRed (rabbit polyclonal) | Takara Bio | Takara Bio Cat# 632496, RRID:AB_10013483 | IHC (1:1000) |
| Antibody | anti-PAX5 (goat polyclonal) | Santa Cruz | Santa Cruz Biotechnology Cat# sc-1974, RRID:AB_2159678 | IHC (1:1000) |
| Antibody | anti-SATB2 (guinea pig polyclonal) | Synaptic Systems | Synaptic Systems Cat# 327 004, RRID:AB_2620070 | IHC (1:1000) |
| Antibody | anti-COUP-TFII (anti-NR2F2, mouse monoclonal) | Perseus Proteomics | Perseus Proteomics Cat# PP-H7147-00, RRID:AB_2314222 | IHC (1:1000) |
| Antibody | anti-VGLUT3 (guinea pig polyclonal) | Synaptic Systems | Synaptic Systems Cat# 135 204, RRID:AB_2619825 | IHC (1:500) |
| Antibody | anti-RFP (rat monoclonal) | Chromotek | ChromoTek Cat# 5f8-100, RRID:AB_2336064 | IHC (1:500) |
| Antibody | anti-DCX (goat polyclonal) | Santa Cruz | Santa Cruz Biotechnology Cat# sc-8066, RRID:AB_2088494 | IHC (1:1000) |
| Antibody | anti-Ki67 (rat monoclonal) | Thermo Fisher Scientific | Thermo Fisher Scientific Cat# 14-5698-80, RRID:AB_10853185 | IHC (1:1000) |
| Antibody | anti-TPH2 (rabbit polyclonal) | Novus Biologicals | Novus Cat# NB100-74555, RRID:AB_1049988 | IHC (1:1000) |
| Antibody | Alexa Fluor 488 (donkey anti-chicken) | Jackson | Jackson ImmunoResearch Labs Cat# 703-545-155, RRID:AB_2340375 | IHC (1:500) |
| Antibody | Alexa Fluor 546 (donkey anti-rabbit) | Thermo Fisher Scientific | Thermo Fisher Scientific Cat# A10040, RRID:AB_2534016 | IHC (1:500) |
| Antibody | Alexa Fluor 647 (donkey anti-goat) | Thermo Fisher Scientific | Thermo Fisher Scientific Cat# A-21447, RRID:AB_2535864 | IHC (1:500) |

*Continued*

| Reagent type (species) or resource | Designation | Source or reference | Identifiers | Additional information |
|---|---|---|---|---|
| Antibody | Alexa Fluor 647 (donkey anti-mouse) | Jackson | Jackson ImmunoResearch Labs Cat# 715-605-151, RRID:AB_2340863 | IHC (1:500) |
| Antibody | Alexa Fluor 647 (donkey anti-rabbit) | ThermoFisher Scientific | Thermo Fisher Scientific Cat# A-31573, RRID:AB_2536183 | IHC (1:500) |
| Antibody | Cy5 (donkey anti-guinea pig) | Jackson | Jackson ImmunoResearch Labs Cat# 706-175-148, RRID:AB_2340462 | IHC (1:500) |
| Commercial assay or kit | *Tph2*-C2 probe | ACDBio | ACDBio:318691-C2 | |
| Commercial assay or kit | Chromium Single Cell v3 Reagent Kits | 10X Genomics | 10X Genomics:1000092/1000074 | |
| Commercial assay or kit | SMARTseq V4 Ultra Low Input RNA Kit | Takara Bio | Takara Bio:634890 | |
| Commercial assay or kit | Nextera XT DNA Library Preparation Kit | Illumina | Illumina: FC-131–1024 | |
| Commercial assay or kit | RNAscope Fluorescent Multiplex Reagent Kit | ACDBio | ACDBio:320850 | |
| Software, algorithm | R (Version 3.5.3, 3.6.3) | R Project for Statistical Computing | R Project for Statistical Computing, RRID:SCR_001905 | https://cran.r-project.org/ |
| Software, algorithm | Seurat (Versions 3.0.2, 3.1.1, 3.1.4) | PMID:29608179 | Seurat, RRID:SCR_016341 | https://satijalab.org/seurat/ |
| Software, algorithm | Fiji (Version 2.0.0-rc-69/1.52 p) | PMID:22743772 | Fiji, RRID:SCR_002285 | https://imagej.net/Fiji |

## Intersectional genetic fate mapping

Triple transgenic mice were generated by crossing *Pet1-Flpe*; RC-FrePe (*Brust et al., 2014*; *Jensen et al., 2008*; *Okaty et al., 2015*) or *Pet1-Flpe*; RC-Ai65 (*Madisen et al., 2015*) mice with *Slc6a4-cre* (*Gong et al., 2007*), *Npy2r-cre* (*Chang et al., 2015*), *En1-cre* (*Kimmel et al., 2000*), *Crh-cre* (https://www.mmrrc.org/catalog/sds.php?mmrrc_id=30850), and *P2ry1-cre* (*Chang et al., 2015*) mice, or by crossing *Pet1-Flpe*; RC-FL-hM3Dq (*Sciolino et al., 2016*) mice with *Slc17a8-cre* mice (https://www.jax.org/strain/028534). All animals were group housed (five animals per ventilated cage) on a 12 hr light/dark cycle with access to food and water ad libitum and were handled and euthanized in accordance with Harvard's Institutional Animal Care and Use Committee Protocols.

## Perfusion and immunohistochemistry

Anesthetized mice were transcardially perfused with cold phosphate-buffered saline (PBS) followed by 4% paraformaldehyde (PFA). Tissue was dissected and fixed in 4% PFA overnight followed by cryoprotection in 30% sucrose/PBS until equilibrated (~48 hr) before being frozen in tissue freezing medium (Triangle Biomedical Services). Tissue was cryosectioned in 40 um coronal sections and processed as floating sections.

For fluorescent staining, sections were washed with PBS and PBS with 0.1% Triton-X-100 (PBS-T), blocked in 5% normal donkey serum (NDS) and 1% bovine serum albumin (BSA) for 2 hr at room temperature (RT), and incubated with primary antibody at 4°C for 48 hr: anti-GFP (1:3000, chicken polyclonal, Aves Labs, GFP-1020), anti-DsRed (1:1000, rabbit polyclonal, Takara, 632496), anti-TPH2 (1:1000, rabbit polyclonal, Novus Biologicals, NB100-74555), anti-Pax5 (1:1000, goat polyclonal, Santa Cruz, sc-1974), anti-SATB2 (1:1000, guinea pig polyclonal, Synaptic Systems, 327–004), anti-COUP-TFII (1:1000, mouse monoclonal, Perseus Proteomics, PP-H7 147–00), anti-ZEB2 (1:200, rabbit polyclonal, MyBioSource, MBS9601451), anti-VGLUT3 (1:500, guinea pig polyclonal, Synaptic Systems, 135–204), anti-RFP (1:500, rat monoclonal, Chromotek, 5f8-100), anti-Doublecortin (1:1000,

goat polyclonal, Santa Cruz, SC-8066), and anti-Ki-67(1:1000, rat monoclonal, Invitrogen, 14-5698-80). For fluorescent detection, sections were washed in PBS-T and incubated with species matched secondary antibodies- Alexa Fluor 488 (donkey anti-chicken, Jackson, 703-545-155), Alexa Fluor 546 (donkey anti-rabbit, Invitrogen, A10040), Alexa Fluor 647 (donkey anti-goat, Invitrogen, A21447 or donkey anti-mouse, Jackson, 715-605-151), and Cy5 (donkey anti-guinea pig, Jackson, 706-175-148)- at 1:500 dilution for two hours. Sections were washed in PBS and 1:3000 DAPI before rinsing and mounting onto slides.

## Confocal and fluorescent microscopy and quantification

### Overview images
Overview images of intersectional subtypes were acquired using a 5x objective on a Zeiss Axioplan2 fluorescence microscope equipped with an Axiocam digital camera and Axiovision software using 1 × 1 binning. Images were then cropped to a 1000 × 1000 pixel square containing the dorsal raphe. Images showing the distribution of PAX5, SATB2, and NR2F2 are 2 × 2 tiled maximum intensity images acquired using a Plan Apo λ 20x/0.75 DIC I objective on a spinning disk confocal. Images showing TPH2 and VGLUT3 staining are a single optical slice taken on a spinning disk confocal using a Plan Apo λ 20x/0.75 DIC I objective or Plan Fluor 40x/1.3 Oil DIC H/N2 objective respectively. Images were cropped to create a zoomed image of the region of interest.

### Flat mount of lateral wall of lateral ventricle
P2ry1-cre; Pet1-Flpe; RC-Ai65 mice (n = 4) were transcardially perfused with cold PBS. Lateral wall dissection was completed as described in *Mirzadeh et al., 2010*. Briefly, brains were dissected into PBS and split into two hemispheres. The hippocampus was removed, exposing the lateral wall, and the brain was fixed overnight in 4% PFA in PBS. The remainder of the microdissection of the lateral wall was then completed and immediately proceeded to immunohistochemistry as described above.

### Quantification of immunofluorescence
Quantification of PAX5, NR2F2, and SATB2 was completed in *Slc17a8-cre; Pet1-Flpe; RC-hM3Dq* animals, where cells expressing both *Slc17a8-cre* and *Pet1-Flpe* express an hM3Dq-mCherry fusion and all other *Pet1+* cells express EGFP. Images were acquired as 2 × 2 tiles as a z-stack (0.9 um step) using a Plan Apo λ 20x/0.75 DIC I objective on a spinning disk confocal and cropped into equally sized non-overlapping subregions (1000 × 1000 pixel) spanning the rostral to caudal extent of the dorsal raphe. Cells were counted positive if antibody staining for the protein of interest overlapped with DAPI staining and was within a DsRed + cell (*Slc17a8-cre; Pet1-Flpe* lineage) or a GFP+ cell (subtractive *Pet1* lineage). All counts were completed in images taken from 2 to 4 animals depending on the brain region. Images used for the quantification of VGLUT3 antibody staining were acquired using a Plan Fluor 40x/1.3 Oil DIC H/N2 objective on a spinning disk confocal on non-overlapping anatomical subdivisions of the dorsal raphe. Cells were counted positive based on the overlap of VGLUT3 antibody staining with mCherry (*Slc17a8-cre; Pet1-Flpe* lineage) or a EGFP (subtractive *Pet1* lineage) staining. In the case of TPH2 quantification, *En1-cre; Pet1-Flpe; RC-FrePe* animals were used (EGFP+ *En1-cre; Pet1-Flpe* intersectional lineage cells). Images were acquired as 2 × 2 tiles as a z-stack (0.9 um step) using a Plan Apo λ 20x/0.75 DIC I objective on a spinning disk confocal and cropped into equally sized non-overlapping subregions (1000 × 1000 pixel) spanning the rostral to caudal extent of the dorsal raphe. Cells were counted positive based on colocalization of TPH2 antibody staining with EGFP. All quantification was performed by an experienced observer blinded to the anatomical region of the image in a minimum of two animals per region.

## TPH2/*Tph2* dual immunofluorescence and RNAscope
Transgenic *En1-cre; Pet1-Flpe; RC-FrePe* mice were briefly anesthetized with isoflurane and immediately perfused intracardially with phosphate buffered saline (PBS) followed by 4% paraformaldehyde (PFA) in PBS. Brains were extracted and fixed for 16 hr in 4% PFA at 4 °C, and were then cryoprotected using 30% sucrose in PBS for 48 hr and subsequently embedded in OCT compound (Tissue-Tek). Coronal sections were cut on a cryostat into PBS at 20 µm thickness, rinsed three times with PBS for 10 min and frozen in cryo-storage solution at −30 °C. The day before RNAscope (ACDBio) procedure, the sections were mounted on slides and dried at room temperature (RT) overnight. Prior

to RNAscope, the slides were heated on a slide warmer at 50 °C for 30 min. Sections were re-fixed with 4% PFA for 15 min at RT, followed by Protease III treatment for 30 min at 40 °C. RNAscope was performed based on the manufacturer's protocol: mouse *Tph2*-C2 probe (ACDbio Cat# 318691-C2) and Amp4 AltA were used for hybridization. After the last wash of RNAscope, the slides were washed briefly with PBS followed by permeabilization and blocking with 0.3% TritonX-100% and 5% Normal Donkey Serum (NDS, Jackson ImmunoResearch) in PBS for 1 hr at RT. Then the sections were incubated with primary antibodies in 0.3% TritonX-100% and 3% NDS in PBS for 16 hr at 4 °C. Primary antibodies: chicken polyclonal anti-GFP (1:1000; GFP-1010; Aves Labs), rabbit polyclonal anti-Tph2 (1:1000; NB100-74555; Novus Biological). Sections were then washed with PBS three times for 10 min and incubated with secondary antibodies for 2 hr at RT. Secondary antibodies: donkey anti-chicken IgG-Alexa Fluor 488 (1:500, Jackson ImmunoResearch Laboratories, Inc), donkey anti-rabbit IgG-Alexa Fluor 647 (1:500, ThermoFisher Scientific.). DAPI (4', 6-diamidino-2-phenylindole) was used for nuclear counterstaining.

## Image acquisition and analysis

Images were collected on a Nikon Ti2 inverted microscope with a Yokowaga CSU-W1 spinning disk with a 50 μm pinhole disk, using a Nikon Plan Apo λ 60x/1.4 NA oil-immersion objective and laser lines at 405, 488, 561, and 640 nm, and captured using an Andor Zyla 4.2 Plus sCMOS monochrome camera and Nikon Elements Acquisition Softwarre AR 5.02. Laser settings were adjusted for each sample but kept constant throughout image collection. A custom Fiji (*Schindelin et al., 2012*) macro script was used to process and analyze these images in a semi-automatic manner (*Senft, 2020*; copy archived at https://github.com/elifesciences-publications/RNAscope-IHC-Coloc-alization-in-ImageJ). Analysis was performed on maximum intensity projections of 5 μm thick z-stacks. To segment cells, the GFP and TPH2 channels were processed with a 100 pixel rolling ball background subtraction to remove uneven background fluorescence and a two pixel gaussian blur to aid in cell segmentation. Cells were segmented first by automatic thresholding using the 'Default' Fiji autothreshold method. Cells in the resulting binary images were separated using the Adjustable Watershed plugin (Michael Schmid, https://imagejdocu.tudor.lu/plugin/segmentation/adjustable_watershed/start), which allows the user to manually adjust the default ImageJ binary watershed tolerance. Next, cells were filtered for size (minimum area 70 $\mu m^2$) and circularity (minimum 0.3) using Fiji's 'Analyze Particles.' Segmented cells were checked manually by a user who could delete or redraw ROIs using the freehand tool and the ROI manager. After segmenting both channels sequentially, 2D overlap-based colocalization of the TPH2 and GFP-labeled cells was performed in which a cell was considered as 'colocalized' if area overlap was greater than 60% of the GFP area (determined empirically). To quantify *Tph2* RNAscope puncta within cells, a 50 pixel rolling ball background subtraction was applied before isolating puncta as local intensity maxima using the Fiji 'Find Maxima' function with a prominence level of 100. RNAscope puncta were then counted within each cell ROI and resulting data were output to a spreadsheet. For GFP+ soma located close to TPH2+ cells such that their outlines in the maximum projection partially overlapped with TPH2 signal but did not reach the colocalization criterion, RNAscope puncta were only counted in the TPH2- region of the cell soma to avoid including puncta belonging to the neighboring cell.

## Stereotaxic surgery

Mice were anesthetized with 1.5–2.0% isofluorane and placed in a stereotaxic system (Kopf). Using a Micro4 injector (WPI) and Nanofil sytringe equipped with a metal 33 g beveled needle (WPI) 75 nL of pENN.AAV.hSyn.Cre.WPRE.hGH was injected at 50 nL/min into the lateral ventricle using using the following stereotaxic coordinates: −0.34 mm AP, + 1.00 mm LM, −2.25 mm DV (AP is relative to bregma). After surgery, mice recovered on a heated pad until ambulatory and returned to their home cage. pENN.AAV.hSyn.Cre.WPRE.hGH was a gift from James M. Wilson (Addgene viral prep #105553-AAVrg; http://n2t.net/addgene:105553; RRID:Addgene_105553).

## Single-cell sorting and RNA sequencing

### On-chip sort, 10X library preparation, and RNA sequencing

Data was derived from two different experiments composed of brain tissue harvested from *En1-cre*; *Pet1-Flpe*; RC-FrePe mice (n = 4) or *Pet1-Flpe*; RC-FL-hM3Dq mice (n = 6). Tissue was sectioned on

a vibratome and protease-digested in ACSF containing activity blockers as described in *Hempel et al., 2007*. The dorsal raphe was micro-dissected under an upright dissection microscope with fluorescence optics and all tissue was combined in a 1.5 mL Eppendorf tube containing 500 ul of filtered ACSF/1%FBS. Tissue was then gently triturated using glass micropipettes of decreasing diameter until achieving a mostly homogeneous single-cell suspension without visible tissue chunks. One drop of NucBlue (Thermo Fisher Scientific) was added to the cell suspension and allowed to sit for 20 min (to aid in sorting and cell quantification). EGFP-marked, NucBlue-positive cells were sorted using the On-chip Sort (On-chip Biotechnologies Co., Ltd.). Final cell concentration was determined by counting the number of cells in 10 ul of the sorted output using a hemacytometer. Cells were then run through the 10X Genomics Chromium Single Cell 3′ v3 protocol, and libraries were sequenced on an Illumina NextSeq 500 sequencer to a mean depth of ~115,000 reads per cell.

## Manual sorting and RNA sequencing

Brain tissue was harvested from triple transgenic animals – *Slc6a4-cre; Pet1-Flpe; RC-FrePe, Npy2r-cre; Pet1-Flpe; RC-FrePe, Crh-cre; Pet1-Flpe; RC-FrePe, and P2ry1-cre; Pet1-Flpe; RC-FrePe* (p60-p120, a minimum of two mice per condition) and fluorescently labeled cells were sorted as described in *Okaty et al., 2015*. Briefly, the brainstem was sectioned into 400 um coronal sections using a vibratome. Sections were bubbled in artificial cerebrospinal fluid (ACSF) containing activity blockers for at least 5 min before being transferred to ACSF containing 1 mg/ml pronase for 1 hr. Slices were then returned to protease-free ACSF for 15 min, before regions of interest were micro-dissected. Anatomical subdivisions of the dorsal raphe were made based on the shape of the dorsal raphe and landmarks including fiber tracts and the aqueduct (as indicated in *Figure 4A*). Dissected chunks of tissue were transferred first to a clean 35 mm dish containing ACSF and then to a 1.5 mL Eppendorf tube containing 1 mL of filtered ACSF/1% FBS. Tissue was then gently triturated until without visible chunks. Dissociated cells were diluted and poured into a Petri dish. Fluorescently marked cells were aspirated using mouth aspiration and moved into three consecutive wash dishes. Each cell was then aspirated a final time and deposited into an individual 0.5 mL tube containing 9.5 ul of nuclease-free water and 1 ul of 10x Reaction Buffer (Smart-Seq V4 Ultra Low Input RNA kit, Takara Bio) and allowed to incubate at room temperature for 5 min before being stored at −80 deg until cDNA synthesis. Single cells were converted to cDNA and amplified using Smart-Seq V4 Ultra Low Input RNA Kit (Takara Bio). The cDNA output was then processed with Nextera XT DNA Library Preparation Kit. Quantification and quality control were assessed with TapeStation. Libraries were then sequenced on either an Illumina HiSeq 2500 (50 base-pair, single-end) or NextSeq 500 (75 bp, paired-end) to a mean depth of ~4,000,000 reads per cell.

## scRNA-seq analysis

### 10x scRNA-seq data

Transcriptome mapping (using the mm10 genome assembly) and demultiplexing were performed using the 10X Genomics Cell Ranger software (version 3.0.2). Several data-filtering steps were performed on the matrix of transcript counts (using R version 3.5.3) prior to further analysis. First, we filtered out all genes detected in fewer than ten single-cell libraries, and filtered out all libraries with less than 4,500 detected genes. This threshold was selected based on the histogram of gene detection for all single-cell libraries as initially called by the Cell Ranger cell detection algorithm, which appeared to reflect two different distributions corresponding to low-complexity versus high-complexity libraries. The low-complexity distribution was right-skewed and had a mode of less than 1,000 detected genes, whereas the high-complexity distribution was left-skewed and had a mode of ~7,500 detected genes. 4,500 genes was roughly the boundary between the two distributions; that is the minima between the two modes, and also corresponded to a sharp inflection point in the Barcodes versus UMI counts plot in the web_summary.html file generated by Cell Ranger. While many of these low-complexity libraries may have been misidentified as cells by Cell Ranger (e.g. droplets containing transcripts from lysed cells, rather than intact cells) examination of genes enriched in lower-complexity libraries suggested that some of them reflected unhealthy cells (e.g. libraries with high mitochondrial gene expression) or contaminating non-neuronal cells (e.g. libraries enriched for glial marker genes). Notably, the number of cells with high-complexity libraries corresponded well with our estimated number of EGFP positive cells used as input to the 10X chip. We

further excluded libraries with: (1) evidence of glial contamination, based on high-outlier expression of glial marker genes, including *Plp1*, *Olig1*, and *Aqp4*, (2) absence or low-outlier levels of *Pet1/Fev* transcripts, (3) greater than fifteen percent of detected genes corresponding with mitochondrial genes, (4) less than two percent of detected genes corresponding with ribosomal genes (these appeared to be single-nuclei libraries, rather than single-cell), (5) high-outlier UMI counts, and (6) high-outlier gene detection. 2,350 single-cell libraries and 17,231 genes passed the above filtering criteria.

Next, we created a Seurat object using these filtered data (Seurat version 3.0.2). Data were log-normalized using the NormalizeData function (using the default scale factor of 1e4), and we identified the top two thousand genes (or in some cases non-coding RNAs) with the most highly variable transcript expression across single cells using the FindVariableFeatures function (selection.method = 'vst', nfeatures = 2000). We then scaled and centered the log-normalized data using the ScaleData function and carried out principal components analysis (PCA) on the scaled expression values of the two thousand most highly variable genes. This allowed us to reduce the dimensionality of the data onto a smaller set of composite variables representing the most salient gene expression differences across single neurons. The procedure for identifying meaningful *Pet1* neuron subtype clusters is thoroughly described in the Results section of the main text. Briefly, we systematically varied the number of principal components included and the resolution parameter in the functions FindNeighbors, FindClusters, and RunUMAP, Dendrograms were created using BuildClusterTree and PlotClusterTree, and cluster-enriched genes were identified using the FindAllMarkers function, with min. pct = 0.25 and logfc.threshold = 0.25, using Wilcoxon Rank Sum tests. This function adjusts p-values for multiple comparisons using the Bonferroni correction. All genes found to be significantly enriched or 'de-enriched' (i.e. expressed at a significantly lower level) in each cluster, as well as the top two thousand highest variance genes, can be found in *Supplementary file 1*.

## Manual scRNA-seq data

Transcript mapping to the mm10 genome assembly and feature counts were performed using STAR (version 2.5.4) (*Dobin and Gingeras, 2016*). Given the high purity of manual cell sorting and the high sensitivity of SMART-Seq v4 cDNA amplification, no data filtering was required; that is single-cell libraries showed no evidence for off-target contamination and showed consistently high gene detection (~9,000 genes per single-cell). Counts data were analyzed using Seurat as described for 10X scRNA-seq data.

## Transfer of 10x cell type labels

In order to explore the correspondence between the fourteen 10X scRNA-seq data-defined *Pet1* neuron subtypes and other scRNA-seq data, including our manual scRNA-seq data, and the *Huang et al., 2019* and *Ren et al., 2019* datasets, we employed the strategy outlined in *Stuart et al., 2019*. Specifically, we used the Seurat functions FindTransferAnchors and Transfer-Data, using the 10X data as the 'reference' and the other datasets as the 'query' group.

## Electrophysiology methods

In vitro brainstem slice preparations containing dorsal raphe serotonin neurons were obtained from 4 to 5 week old mice. After isofluorane anesthesia, mice were perfused transcardially with a solution of artificial CSF (NaHCO$_3$-aCSF) containing the following (in mM): 124 NaCl, 25 NaHCO3, 3 KCl, 2 CaCl2, 2 MgCl$_2$, 1.2, NaH2PO4 and 25 d-Glucose, equilibrated with 95% O2% and 5% CO2 adjusted to 310 ± 5 mOsm/L. The brainstem was dissected and mounted on the stage of a VT1200S vibratome while immersed in an ice slush solution aCSF containing the following (in mM): NMDG 93, HCl 93, KCL 2.5, NaH$_2$P0$_4$ 1.2, NaHCO$_3$ 30, HEPES 20, d-Glucose 25, Na-Ascorbate 5, Thiourea 2, Na-Pyruvate 3, MgSO$_4$ 10, CaCl$_2$ 0.5 equilibrated with 95% O2% and 5% CO2 adjusted to 310 ± 5 mOsm/L. Coronal slices 200 μm thick containing the dorsal nucleus raphe were recovered for 1 hr at 35–6 C in HEPES-aCSF containing: NaCl 92, KCl 2.5, NaH$_2$PO$_4$ 1.2, NaHCO$_3$ 30, HEPES 20, Glucose 25, Na-Ascorbate 5, Thiourea 2, NaPyruvate 3, MgSO$_4$ 10, CaCl$_2$ 0.5 equilibrated with 95% O2% and 5% CO2 adjusted to 310 ± 5 mOsm/L and placed at room temperature for storage. Individual slices were transferred to the recording chamber and superfused with NaHCO3-aCSF at 34°C. Electrodes (5–7 MΩ) were pulled from borosilicate glass. Pipettes were filled with (in mM): 140

K-gluconate, HEPES 10, KCl 5, Na-ATP 2, MgCl$_2$ 2, EGTA 0.02, biocytin 0.1% Na$_2$GTP 0.5, Na$_2$-phosphocreatine 4, pH 7.4 adjusted with KOH and adjusted to 285 ± 5 mOsm/L with sucrose. Somatic whole-cell recordings were obtained with a Multiclamp 700B amplifier, signals were acquired and sampled at 100 kHz using Digidata 1440A digitizing board. Pipette capacitance was compensated ≈70% in current clamp (CC). Series resistance ($R$s) was typically 9–15 MΩ. Cells with $R$s >15 MΩ were discarded. A measured liquid junction potential of ≈10 mV was corrected online. Cells were held at V$_h$ = −80 mV unless otherwise indicated. To create action potential frequency-current curves, a protocol that applies a series of 750 ms current pulses ranging from −100 pA to 220 pA was created using Molecular Devices Clampex 10.7 software running on Windows 7.

## Acknowledgements

The authors thank the Biopolymers Facility at HMS for assistance with next-generation sequencing; the Microscopy Resources on the North Quad (MicRoN) core at Harvard Medical School for microscope use and training; Steve Liberles for providing *Npy2r-cre* and *P2ry1-cre* driver lines; ChangHee Lee and Jin Akagi for advice with the On-chip Sort; Kathryn Commons and the Dymecki lab for discussions and thoughtful comments on this manuscript; J.J Mai for reagents and animal husbandry. Grants supporting this work include NARSAD Young Investigator Grant (BWO), NIDA Grant RO1DA034022 (BWO, OVA, SMD), T32 HL 007901 (NS), NIH F31NS108406 (RAS), Howard Hughes Medical Institute Gilliam Fellowship (KAL), NIH F99-NS108515 (KAL), Harvard Brain Science Initiative Bipolar Disorder Seed Grant, supported by Kent and Liz Dauten (OVA), and the GVR Khodadad Fund for the Study of Genetic, Neurobiological, and Physiochemical Processes of EPS (OVA, BWO, SMD).

## Additional information

### Funding

| Funder | Grant reference number | Author |
| --- | --- | --- |
| NARSAD Young Investigator Grant | 27594 | Benjamin W Okaty |
| National Institute on Drug Abuse | RO1DA034022 | Benjamin W Okaty<br>Olga V Alekseyenko<br>Susan M Dymecki |
| National Heart, Lung, and Blood Institute | T32HL007901 | Nikita Sturrock |
| GVR Khodadad Fund for the Study of Genetic, Neurobiological, and Physiochemical Processes of EPS | | Benjamin W Okaty<br>Olga V Alekseyenko<br>Susan M Dymecki |
| National Institute of Neurological Disorders and Stroke | NIH F31NS108406 | Rebecca A Senft |
| Howard Hughes Medical Institute Gilliam Fellowship | | Krissy A Lyon |
| Harvard Brain Science Initiative Bipolar Disorder Seed Grant, supported by Kent and Liz Dauten | | Olga V Alekseyenko |
| NIH Blueprint for Neuroscience Research | F99 NS108515 | Krissy A Lyon |

The funders had no role in study design, data collection and interpretation, or the decision to submit the work for publication.

### Author contributions

Benjamin W Okaty, Conceptualization, Data curation, Software, Formal analysis, Supervision, Funding acquisition, Validation, Investigation, Visualization, Methodology, Writing - original draft, Project

administration, Writing - review and editing; Nikita Sturrock, Conceptualization, Data curation, Formal analysis, Funding acquisition, Validation, Investigation, Visualization, Methodology, Writing - original draft, Project administration, Writing - review and editing; Yasmin Escobedo Lozoya, Formal analysis, Investigation, Visualization, Methodology, Writing - review and editing; YoonJeung Chang, Krissy A Lyon, Olga V Alekseyenko, Investigation; Rebecca A Senft, Software, Formal analysis, Methodology; Susan M Dymecki, Conceptualization, Supervision, Funding acquisition, Writing - original draft, Project administration, Writing - review and editing

### Author ORCIDs
Benjamin W Okaty (iD) https://orcid.org/0000-0003-1281-2244
Nikita Sturrock (iD) https://orcid.org/0000-0002-1635-6760
Yasmin Escobedo Lozoya (iD) https://orcid.org/0000-0001-8197-770X
YoonJeung Chang (iD) https://orcid.org/0000-0001-9549-8208
Rebecca A Senft (iD) https://orcid.org/0000-0003-0081-4170
Krissy A Lyon (iD) https://orcid.org/0000-0002-4453-8406
Olga V Alekseyenko (iD) https://orcid.org/0000-0003-1645-5133
Susan M Dymecki (iD) https://orcid.org/0000-0003-0910-9881

### Ethics
Animal experimentation: Procedures were in accordance with institutional animal care and use committee (IACUC) policies at Harvard Medical School, specifically as outlined and approved in IACUC protocol IS00000231.

### Decision letter and Author response
Decision letter https://doi.org/10.7554/eLife.55523.sa1
Author response https://doi.org/10.7554/eLife.55523.sa2

## Additional files
### Supplementary files
• Supplementary file 1. The 'all_subgroup_markers' worksheet displays the output of the Seurat FindAllMarkers function. Column one is the gene symbol, column two is the p-value given by the Wilcoxon Rank Sum test, column three is the average 'log fold change' (i.e. log-fold difference in transcript abundance between the in-group and out-group), where a positive value indicates that a gene is expressed at a higher level in a given cluster relative to all other clusters, and a negative value indicates that a gene is expressed at a lower level. Column four is the percent of cells within a particular cluster in which the gene was detected, column five is the percent of cells within all other clusters in which a gene was detected, column six gives the Bonferroni-corrected p-value, and column seven indicates the cluster in which the given gene is a positive or negative marker. Note, not all enriched genes are unique to only one cluster, as more similar clusters will share subsets of enriched genes. The 'sig_var_genes' worksheet lists the top two thousand highest 'standardized variance' genes, that is genes that vary significantly more than expected based on mean expression.

• Transparent reporting form

### Data availability
The RNA-seq dataset has been deposited to GEO under the accession number GSE144980.

The following dataset was generated:

| Author(s) | Year | Dataset title | Dataset URL | Database and Identifier |
|---|---|---|---|---|
| Okaty BW, Sturrock N, Escobedo Lozoya Y, Chang Y, Senft RA, Lyon KA, Alekseyenko OV, | 2020 | A single-cell transcriptomic and anatomic atlas of mouse dorsal raphe Pet1 neurons | https://www.ncbi.nlm.nih.gov/geo/query/acc.cgi?acc=GSE144980 | NCBI Gene Expression Omnibus, GSE144980 |

Dymecki SM

The following previously published datasets were used:

| Author(s) | Year | Dataset title | Dataset URL | Database and Identifier |
|---|---|---|---|---|
| Niederkofler V, Asher TE, Okaty BW, Rood BD | 2016 | Intersectionally labeled Drd2-Pet1 single-neuron RNA-seq | https://www.ncbi.nlm.nih.gov/geo/query/acc.cgi?acc=GSE87758 | NCBI Gene Expression Omnibus, GSE87758 |
| Ren J, Isakova A, Friedmann D, Zeng J | 2019 | Single-Cell Transcriptomes and Whole-Brain Projections of Serotonin Neurons in the Mouse Dorsal and Median Raphe Nuclei | https://www.ncbi.nlm.nih.gov/geo/query/acc.cgi?acc=GSE135132 | NCBI Gene Expression Omnibus, GSE135132 |
| Huang KW, Ochandarena NE, Philson AC, Hyun M | 2019 | Molecular and anatomical organization of the dorsal raphe nucleus | https://www.ncbi.nlm.nih.gov/geo/query/acc.cgi?acc=GSE134163 | NCBI Gene Expression Omnibus, GSE134163 |

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
