## [Decision Letter]

**Acceptance summary:**

The authors used a combination of scRNAseq and intersectional genetic labeling with axon tracing to identify molecular and functional diversity within the population of serotonergic neurons. The study presents a particularly thoughtful use of single-cell sequencing analysis that will be of broad utility to other researchers using this method. The data will serve as an important reference for future studies of serotonergic cell function as well as the understanding of the developmental basis of this diversity.

**Decision letter after peer review:**

Thank you for submitting your article "A single-cell transcriptomic and anatomic atlas of mouse dorsal raphe *Pet1* neurons" for consideration by *eLife*. Your article has been reviewed by three peer reviewers, one of whom is a member of our Board of Reviewing Editors, and the evaluation has been overseen by Kate Wassum as the Senior Editor. The following individuals involved in review of your submission have agreed to reveal their identity: Evan Denerisn (Reviewer #2) and Thomas Perlmann (Reviewer #3).

The reviewers have discussed the reviews with one another and the Reviewing Editor has drafted this decision to help you prepare a revised submission.

Summary:

In this work, Okaty et al. have used a combination of scRNAseq and intersectional genetic labeling to characterize neurons with a history of Pet-1 expression. The project has resulted in an impressive molecular characterization of 5-HT neurons uncovering 14 different subgroups of neurons. Bioinformatic analysis has allowed identification of subgroup-specific gene expression patterns and the combination with intersectional genetic studies and histology has resulted in further anatomical description also complemented with electrophysiology of selected 5-HT neuron types. The study is well performed and the data will serve as a good reference for future studies of serotonergic cell function as well as the understanding of the developmental basis of this diversity.

The reviewers were impressed with the quality of the work and with the exception of a missing figure panel have only relatively minor concerns that they would like to see addressed.

Essential revisions:

1) Figure 4C is missing. In the subsection “Manual scRNA-seq of *Pet1*-Intersectionally Defined Neuron Populations”, the authors describe how they were able to infer the anatomical distributions of all fourteen clusters and refer to Figure 4C. This figure, however, is not included in the manuscript (Figure 4 contains only A and B). The authors should show some validation in tissue of all 14 subgroups. It would also be helpful for the reader to see a summary figure illustrating how the subgroups are located in a schematic drawing (omitted Figure 4C?).

2) The authors should discuss how sampling may have affected the quality of the data. It is important to describe sampling efficiency (i.e. how big proportion of 5-HT cells can be isolated per brain). This will give the readers an idea of likely sampling bias resulting in over- or under-representation of specific Pet-1 subtypes. Also, it would be interesting to know if the authors have made any attempt to determine if the sequenced cells stoichiometrically correspond to in vivo proportions of different subgroups, which in itself gives an indication of sampling bias. Additional experiments are not needed here, but a careful discussion is warranted. If the total number of recovered neurons is low, there is a bigger risk that subgroups could be severely underrepresented or even lost from analysis. This may also, at least partly, explain differences between this study and the studies by Huang et al. and Ren et al. as they have used different strategies to dissociate and isolate cells.

3) For Figure 5, the authors trace Td-Tomato+ axons in a *Pet1-Flp*/*P2ry1-Cre* double transgenic mouse and make some novel observations about the trajectories of the axons. The authors make a convincing argument that the intersectional strategy is far superior to single transgenes for labeling specific populations of neurons. However it is unlikely to be perfect. Ideally the authors would find a way to trace some of their axons back to cell bodies and use a second method to confirm the identity of these cells. Otherwise, commentary on the possibility of that the cells projecting along novel paths could be something other than a unique serotonin neuron population would be useful.

4) Also with respect to the projection pattern of cluster 12 (*Met-Vglut3-Tph2-Pet1*) neurons, it remains unclear to the reader whether this innervation pattern is unique to this sub-cluster – i.e. do other DR serotonergic neurons project to these same areas? Although it is not needed to investigate the projection patterns of all 13 clusters it would be interesting to know if the authors can detect other non-labelled, serotonergic fibers near ventricles (e.g. using 5-HT immunostaining), or whether all the fibers are indeed originating from cluster 12. In addition, it would be helpful to include an immunostaining of the *Met* cluster itself (using the genetic labelling) in the same image – not only depict it in the schematic figure.

---

## [Author Response]

Essential revisions:1) Figure 4C is missing. In the subsection “Manual scRNA-seq of Pet1-Intersectionally Defined Neuron Populations”, the authors describe how they were able to infer the anatomical distributions of all fourteen clusters and refer to Figure 4C. This figure, however, is not included in the manuscript (Figure 4 contains only A and B). The authors should show some validation in tissue of all 14 subgroups. It would also be helpful for the reader to see a summary figure illustrating how the subgroups are located in a schematic drawing (omitted Figure 4C?).

Figure 4C was referenced in error – it was a figure panel from an earlier draft of the manuscript that we ultimately chose to remove. The reviewers have correctly surmised that Figure 4C was a schematic drawing of subgroup distributions, however we decided that the relationships between *Pet1* neuron molecular subgroups and DR anatomy revealed by our data were more informatively conveyed in the summary table in Figure 8. We now include a schematic drawing to the revised Figure 8 to complement the table.

To address the reviewer requests for validation of subgroup anatomy, we have added new immunohistochemistry and in situ hybridization data and revised the manuscript to more explicitly state how we used our data, as well as links between our data and other datasets, to infer certain subgroup distributions. Specifically we have added a new figure summarizing all the evidence that went into our imputed anatomical mappings (new Figure 8—source data 1), which is a combination of: (1) histological analyses of intersectional mouse lines (Figure 3), (2) immunohistology (including antibodies against TPH2, VGLUT3, PAX5, SATB2, and NR2F2; Figure 3—figure supplement 1 and 2, Figure 4—figure supplement 2), (3) anatomically targeted scRNAseq (Figure 4), (4) dual TPH2 immuno and *Tph2* single molecule fluorescent in situ hybridization newly added in this revised manuscript (new Figure 3—figure supplement 1), (5) computational mappings of our subtypes to other datasets that performed their own anatomical validation (Figure 7), and (6) qualitative (by eye) analysis of anatomical expression profiles (in situ hybridization) of subtype marker genes represented in the Allen Brain Atlas (as indicated in Figure 8—source data 1).

2) The authors should discuss how sampling may have affected the quality of the data. It is important to describe sampling efficiency (i.e. how big proportion of 5-HT cells can be isolated per brain). This will give the readers an idea of likely sampling bias resulting in over- or under-representation of specific Pet-1 subtypes. Also, it would be interesting to know if the authors have made any attempt to determine if the sequenced cells stoichiometrically correspond to in vivo proportions of different subgroups, which in itself gives an indication of sampling bias. Additional experiments are not needed here, but a careful discussion is warranted. If the total number of recovered neurons is low, there is a bigger risk that subgroups could be severely underrepresented or even lost from analysis. This may also, at least partly, explain differences between this study and the studies by Huang et al. and Ren et al. as they have used different strategies to dissociate and isolate cells.

For our 10X scRNA-seq experiments, we profiled 2,350 individual *Pet1* neurons, which were purified from a total of ten mouse brains (using the On-chip sort). This comes to about 235 cells per microdissected dorsal raphe on average, and in aggregate represents a roughly 25% sampling of the total population of DR 5-HT neurons in a single mouse brain based on the Ishimura, et al., 1988 estimate of ~9,000 5-HT immunoreactive DR cells (though not all DR *Pet1* neurons are 5-HT immunoreactive, and likewise roughly six percent of our *Pet1* cells expressed significantly lower levels of *Tph2* transcripts than the majority). By comparison, the Huang et al., 2019 and Ren, et al., 2019 studies profiled fewer DR 5-HT and *Pet1* neurons than our study. Huang, et al. sequenced a total of 704 DR 5-HT neurons (classified post-hoc by clustering and marker gene expression) and 29 putative non-5HT DR *Pet1-Slc17a8* neurons. Ren, et al. sequenced a total of 567 DR 5-HT neurons, as well as 432 putative MR 5-HT neurons (including 105 neurons that bear some similarities to a caudal and ventromedial DR *Pet1* neuron subtype identified in our study, suggesting potential anatomical overlap as speculated on in our manuscript). Thus, if we just focus on “*Tph2*^high^” DR *Pet1* neurons (Ren, et al. did not target putative non-5-HT *Pet1* neurons), we sampled more than three times as many neurons as both Huang, et al. and Ren, et al. These differences in the numbers of profiled neurons are likely the main reason more subgroups were identified here than the other two studies. Other differences between studies, such as differences in sensitivity of gene detection and different cell targeting and library preparation strategies (e.g. InDrops of non-sorted DR cells (Huang, et al.,) versus sorting of *Pet1* (here) or *Slc6a4* enhancer-driven fluorescently marked neurons (Ren et al.,) followed by 10X or SMARTseq library preparation protocols), also likely influenced subtype classification.

Related to these points, we have revised the manuscript to include the results of new analyses we performed to explicitly model how the number of cells sampled and the number of UMIs identified per cell influence the number of identified *Pet1* neuron subgroup clusters (Figure 7—figure supplement 1, described in the subsection “Comparison to other DR scRNA-seq dataset”). Specifically, we randomly sub-sampled variable numbers of cells (twenty times per iteration) and re-clustered the resulting data in the same manner described for the full dataset (using a Seurat FindClusters resolution of 0.9), varying the number of sampled cells from 200 to 2,300 (new Figure 7—figure supplement 1A). As expected, we found that increasing the number of cells increased the number of clusters. Identification of fourteen clusters did not occur until at least 1,700 cells were included in the analysis, and this number of clusters began to stabilize as 2,100 or more cells were included. Similarly, we randomly sub-sampled UMIs, varying the maximum number of UMIs per cell from 500 to 100,000, and repeated our clustering analysis twenty times per subsampled max UMI (new Figure 7—figure supplement 1B). In this case we found a much steeper relationship between max UMIs and number of clusters identified, with fourteen clusters being identified with as few as 4,500 max UMIs per cell, and completely stabilizing at roughly 60,000 max UMIs. With respect to our ability to resolve fine-scale DR *Pet1* neuron subgroup structure, these results indicate that while both variables are important, the number of cells sampled was to some extent “more limiting” than the number of UMIs in our dataset; i.e. we could have uncovered a similar degree of overall cellular diversity (fourteen subtypes) with less “complex” libraries (e.g. from more shallow sequencing), however we needed nearly 90% of the cells we sampled to consistently uncover fourteen molecular subgroups.

The results of these analyses shed light on the most likely reasons why we were able to achieve more fine-grained classification of DR *Pet1* neuron subtypes. For example, Ren et al. had a similar degree of library complexity to ours, however as noted above they profiled fewer cells – 567 to our 2350 cells (~2,200 excluding *Tph2-low* cells). When we sub-sample our data to a similar number of cells, we find between six and nine clusters, and similarly, Ren, et al. reported seven DR 5-HT neuron clusters. When we simultaneously sub-sample both the number of cells profiled and the maximum number of UMIs detected per single cell to levels similar to the Huang, et al. study (750 cells and 2,500 max UMIs, Figure 7—figure supplement 1C), we uncover between four and seven subgroup clusters – Huang, et al. found six. Thus, all other methodological sources of variation between studies aside, these two parameters plausibly explain differences in the degree of diversity uncovered across studies.

The question of subtype “stoichiometry” posed by the reviewers is indeed interesting, but more difficult to answer. Ideally one would employ spatial transcriptomics or something like MERFISH (Moffitt, et al., 2018) or SABER (Kishi et al., 2019) to address this question, and this is something we will likely pursue in the future. However, one way to get at this at present is to compare the ratios of similar DR *Pet1* neuron subgroups identified by us and by other studies.

For instance, our subgroups 2-6 comprise roughly 53% of DR *Pet1* 5-HT neurons (i.e. *Pet1 Tph2-high* neurons) identified in our study, and taken as a whole they roughly correspond to groups 1-3 from Ren, et al. and Huang, et al. studies, comprising ~44% and ~69% of their respective datasets. Our subgroups 7-11 and 14 collectively make up ~37% of all *Pet1* 5-HT neurons profiled, and bear similarity to Ren, et al. DR 4-5 and Huang, et al. group 4, making up ~35% and ~30% of those datasets, respectively. Our subgroup 12 comprises ~5% of our 5-HT *Pet1* dataset, and it maps precisely to the Ren, et al. cDR group and the Huang, et al. group 5, making up ~4% and 1% of sampled 5-HT neurons in those studies. Thus, subgroup percentages are somewhat similar across studies, but in general our percentages are more similar to the Ren, et al. study than to the Huang, et al. study. However, one pronounced difference in percentage composition between our data and Ren, et al. data is our group 1, which maps well to their DR-6 group (Huang, et al. lacks such a group). We find that ~5 % of our DR *Pet1* 5-HT neurons belong to subgroup cluster 1, whereas ~17% of Ren, et al. DR 5-HT neurons belong to their group 6. Given that these neurons show a ventrolateral bias – in some cases much more lateral than the main cluster of DR neurons – it’s possible that sampling of these neurons would be more sensitive to differences in the boundaries of microdissection.

In terms of the percentage of *Pet1* neurons with significantly lower levels of *Tph2* transcript, our newly added dual TPH2 immuno / *Tph2* mRNA in situ experiments in *En1-cre; Pet1-Flpe; RCFrePe* mice (new Figure 3—figure supplement 1, described in the subsection “Histology of *Pet1*-Intersectionally Defined Neuron Populations”) allow us to directly compare scRNA-seq data to “in vivo” proportions. We found that between twenty and thirty percent of GFP-positive neurons (i.e. neurons with a history of *En1cre* and *Pet1-Flpe* expression) were TPH2-immunonegative, and in general expressed significantly lower levels of *Tph2* transcript (quantified using an in-house automated ImageJ macro described in our Materials and methods). However, *Pet1 Tph2-low* transcript neurons as identified by 10X scRNA-seq (cluster 13) only make up ~6% of all profiled *Pet1* neurons, similar to Huang, et al., where they make up ~4% of *Pet1* neurons. We see two potential reasons for this disparity. First, we found that GFP-positive TPH2-immunonegative neurons were significantly smaller in soma size than double positive GFP-positive TPH2-immunopositive neurons, which may lead to a selection bias against them when cell sorting. Second, as part of our data-filtering process prior to clustering analysis, we required a certain level of *Pet1* transcript detection (described in our Materials and methods), as well as a certain level of overall gene detection (>4500 genes). Given that *Pet1-Flpe* recombination is a permanent event, it’s possible that some GFP positive neurons (for example, in *En1-cre; Pet1-Flpe; RC-FrePe* mice) do not express high enough levels of *Pet1* transcripts at the time of dissociation and profiling (but did earlier in development, or at least enough to express sufficient Flpe to mediate recombination and thus allow GFP expression), and these neurons would therefore have been omitted from our analysis. Consistent with this line of reasoning, we found that removing a minimum *Pet1* transcript detection threshold led to an increase in the number of cluster 13 neurons, going from ~6% to ~9% of all profiled cells. Additionally, it has been noted that cell size may affect library complexity (Wagner, Regev, and Yosef, 2016), with smaller cells potentially exhibiting less complexity. If this is the case, some smaller *Tph2-low* cell libraries may likewise not have passed our filtering thresholds due to lower gene detection.

3) For Figure 5, the authors trace Td-Tomato+ axons in a Pet1-Flp/P2ry1-Cre double transgenic mouse and make some novel observations about the trajectories of the axons. The authors make a convincing argument that the intersectional strategy is far superior to single transgenes for labeling specific populations of neurons. However it is unlikely to be perfect. Ideally the authors would find a way to trace some of their axons back to cell bodies and use a second method to confirm the identity of these cells. Otherwise, commentary on the possibility of that the cells projecting along novel paths could be something other than a unique serotonin neuron population would be useful.

Axons were traced using the intersectional expression of a fluorescent reporter, eGFP or

TdTomato, in *P2ry1-cre; Pet1-Flpe* animals (either triple transgenic *P2ry1-cre; Pet1-Flpe; RCFrePe* mice or triple transgenic *P2ry1-cre; Pet1-Flpe; Ai65* mice). This technique, while allowing for specificity based on intersectional gene expression, cannot be further restricted based on anatomy. As described in –the subsection “cDR P2ry1-cre; *Pet1-Flpe* neurons display unique hodological and electrophysiological properties”, the vast majority of *P2ry1-cre; Pet1-Flpe* cell bodies reside in the caudal dorsal raphe, but a few are additionally found in parts of the rostral dorsal raphe, median raphe, and medullary *Pet1* regions and thus axons could be arising from any of these cells. Furthermore, other *Pet1* populations project to many of the regions mentioned. To gain insight into which *P2ry1-cre; Pet1-Flpe* neurons contribute to specific target regions, we focused on the ventricles, which are the predominant and most unique aspect of the projection pattern for this population. A retrograde hSyn-Cre virus was unilaterally injected into the lateral ventricle of *Pet1-Flpe* reporter mice (shown in a new figure, Figure 5—figure supplement 1), the methods to which are in the subsection “Stereotaxic Surgery”. As described in the subsection “cDR *P2ry1-cre*; *Pet1-Flpe* neurons display unique hodological and electrophysiological properties”, the predominant population of labeled neurons from these experiments was found in the caudal dorsal raphe, just under the aqueduct in bilateral columns. However there were also labeled cells in the median raphe, though far fewer, which has been shown in previous studies. This suggests that at least for the ependymal projections, ventricular axons characterized from the *P2ry1-cre; Pet1Flpe* population are likely arising from the caudal dorsal raphe group.

4) Also with respect to the projection pattern of cluster 12 (Met-Vglut3-Tph2-Pet1) neurons, it remains unclear to the reader whether this innervation pattern is unique to this sub-cluster – i.e. do other DR serotonergic neurons project to these same areas? Although it is not needed to investigate the projection patterns of all 13 clusters it would be interesting to know if the authors can detect other non-labelled, serotonergic fibers near ventricles (e.g. using 5-HT immunostaining), or whether all the fibers are indeed originating from cluster 12. In addition, it would be helpful to include an immunostaining of the Met cluster itself (using the genetic labelling) in the same image – not only depict it in the schematic figure.

We agree that it would be interesting to determine if *P2ry1-cre; Pet1-Flpe* neurons, which represent cluster 12, are the sole population of *Pet1* neurons projecting through the ventricles. As described in our response to comment 3, we performed stereotaxic injections of a retrograde *cre* virus into the lateral ventricle of *Pet1-Flpe; RC-FrePe* mice or *Pet1-Flpe; RC-Ai65*. This would label *Pet1* neurons with supra-ependymal projections. The major labeled population was found in the caudal dorsal raphe in the same anatomical region as *P2ry1-cre; Pet1-Flpe* neurons. However, there were a few cell bodies in the median raphe. This suggests that there are some other populations of *Pet1* neurons projecting to the ventricles. To address the proposed addition of immunostaining for *Met*, we feel that *Met* expression in *P2ry1-cre; Pet1Flpe* neurons was confirmed by manual scRNAseq, as cells sorted from this intersectionally targeted population had the same *Met* expression pattern as those cells in cluster 12 (and are classified as type 12 when mapped onto the 10X data in Figure 4). Furthermore, the expression of *Met* in the caudal dorsal raphe has previously been confirmed by both immunohistochemistry and in-situ (Okaty et al., 2015, Kast et al., 2017, Ren et al., 2019).